# Preference for and resistance to a toxic sulfur volatile opens up a unique niche in *Drosophila busckii*

Venkatesh Pal Mahadevan [1,2], Diego Galagovsky [1], Markus Knaden [1,2] & Bill S. Hansson [1,2] ✉

The ability to tolerate otherwise toxic compounds can open up unique niches in nature. Among drosophilid flies, few examples of such adaptations are known and those which are known are typically from highly host-specific species. Here we show that the human commensal species *Drosophila busckii* uses dimethyldisulfide (DMDS) as a key mediator in its host selection. Despite DMDS's neurotoxic properties, *D. busckii* has evolved tolerance towards high concentrations and uses the compound as an olfactory cue to pinpoint food and oviposition sites. This adaptability is likely linked to insensitivity of the enzyme complex cytochrome c oxidase (COX), which is a DMDS target in other insects. Our findings position *D. busckii* as a potential model for studying resistance to toxic gases affecting COX and offers insight into evolutionary adaptations within specific ecological contexts.

The occupation of novel ecological niches plays a pivotal role in driving speciation[1]. Among the diverse species in the genus *Drosophila*, successful niche specializations often coincide with evolutionary changes in diet and odor coding for niche-specific odorants[2–8]. The wide-ranging diversity of drosophilid niches, encompassing factors such as geographical distribution (ranging from tropical to arctic regions) and host specialization width (ranging from generalists to specialists on a single host), makes them an excellent model system for studying olfactory evolution. However, despite the presence of more than 1000 *Drosophila* species spanning multiple subgenera, only a handful have been thoroughly investigated regarding their life history[9,10].

*Drosophila melanogaster* (subgenus *Sophophora*), commonly known as the vinegar fly, has served as a prominent model species for extensive research into its olfactory neuroecology[3,11–13]. Over the past two decades, several studies have also delved into the ecological niches and evolutionary shifts employed by other drosophilid species, predominantly from the subgenera *Sophophora*, *Drosophila*, and the genus *Scaptomyza*[2,5–8,14–17]. Among these, two species have garnered special attention due to their ability to thrive on toxic hosts. One notable example is *D. sechellia*, which inhabits the Seychelles Islands

archipelago and specializes in feeding on the Noni fruit (*Morinda citrifolia*), known for its toxicity to other drosophilids[18]. *D. sechellia* exhibits crucial physiological adaptations as a specialist on this otherwise toxic fruit. It has also developed olfactory adaptations, including specific tuning of olfactory receptors and increased sensitivity to specific odorants emitted by the Noni fruit[5,19]. The second interesting species is *Scaptomyza flava*, an herbivorous drosophilid that acts as a leaf miner during its larval stages. It has evolved to specialize in plants from the *Brassicaceae* family, a group of plants known to contain toxic glucosinolates[14]. *S. flava* has developed the ability to detoxify these toxic compounds and can detect airborne isothiocyanate signals using a dedicated class of OSNs[14,17]. However, the aforementioned examples are drosophilids restricted either by their geographical location and/or limited to being specialists on a single host. So far, an example of a generalist, cosmopolitan drosophilid species adapted to several toxic hosts remains unknown.

A few reports from the literature have hinted at the possibility of *D. busckii* (*Dbus*) being one such species but with sparse information available about its ancestral origin and natural distribution[20]. The species is considered part of the cosmopolitan guild of *Drosophila*, along with five other *Drosophila* species, and is known to be associated

[1]Department of Evolutionary Neuroethology, Max Planck Institute for Chemical Ecology, Jena, Germany. [2]Max Planck Center next Generation Insect Chemical Ecology, Max Planck Institute for Chemical Ecology, Jena, Germany. ✉e-mail: hansson@ice.mpg.de

with humans in present times, behaving as a commensal[21,22]. The species can be found on various substrates, including rotting vegetables like potatoes, chicory, and mushrooms[20,23–25]. Interestingly, many of the reported breeding hosts for *Dbus*, such as rotting cauliflowers or Brussels sprouts, belong to the cruciferous vegetable family, known for containing high levels of defense compounds perceived as toxic by many insects[26]. Additionally, *Dbus* has been observed to have associations with pathogenic microbe species that are harmful to plants and reported to be involved in causing soft rot in tomatoes (*Aspergillus niger*) and chicory (*Erwinia carotovora*)[25,27]. The existing literature thus suggests an intriguing link between *Dbus* and potentially toxic hosts, making it an excellent candidate for investigating a cosmopolitan, generalist human commensal drosophilid that might have evolved a preference for several toxic hosts.

Here we show that *Dbus* flies display a clear preference for several rotting vegetable and mushroom substrates that emit short-chain oligosulfides, and a specific affinity towards dimethyldisulfide (DMDS), a compound commonly used as a commercial fumigant and known to possess neurotoxic properties[28,29]. *Dbus* also successfully completes its life cycle on these DMDS-emitting substrates. Furthermore, we reveal a specific class of antennal olfactory sensory neurons (OSNs) tuned to detect short-chain oligosulphides, particularly DMDS, indicating a specialized olfactory adaptation in the species.

Next, we show that *Dbus* has developed an impressive ability to tolerate DMDS concentrations that are highly toxic to five other cosmopolitan and co-occurring *Drosophila* species[21,22]. Previous research has established that DMDS exerts its neurotoxic effects by interacting with the mitochondrial cytochrome C oxidase (COX) enzyme, resulting in the inhibition of ATP generation[28]. Our results hint towards the involvement of COX in conferring DMDS resistance while we hypothesize that *Dbus* very likely possesses an insensitive form of COX, which allows the flies to tolerate DMDS.

## Results

We studied the olfactory neuroecology of *Dbus*, representing the subgenus *Dorsilopha* with a possible origin in southeast Asia[30] (Fig. 1a and Supplementary Fig. 5h). Our first objective was to investigate the species' oviposition preference, to identify the most suitable oviposition substrate and to compare it with the model *D. melanogaster* (*Dmel*). To achieve this, we tested eleven different rotting substrates, where *Dbus* had been reported[25,27,31] (Fig. 1b). To minimize variation in the rotting stage of each substrate, we followed a substrate rotting protocol (see "Methods" section, Supplementary Fig. 1a). Moreover, to ensure the presentation of only olfactory stimuli a filter paper covered the rotting substrate in all experiments ensuring no contact between flies and the stimuli (see "Methods" section). Therefore, the term "oviposition on the substrate" refers to "oviposition on agarose plates as a result of stimulation by rotting substrate volatiles" throughout the subsequent text.

In a no-choice assay, *Dbus* laid significantly more eggs on multiple substrates compared to the control (10 µl of distilled water), except for rotting potato, cucumber, and strawberry (Fig. 1b). *Dmel*, on the other hand, retained eggs for up to 48 h when challenged with multiple substrates but exhibited a preference for rotting strawberry, tomato, orange, and surprisingly, onion (Fig. 1b). Comparing egg numbers between *Dmel* and *Dbus* revealed highly significant differences for several substrates (Fig. 1b), indicating a distinct shift in preferred oviposition substrates between the two species. As rotting orange is known to be a preferred oviposition substrate for *Dmel*[32], we conducted a two-choice experiment to examine *Dmel*'s oviposition preference between rotting orange and another rotting substrate (Fig. 1c). The results showed that *Dmel* significantly and consistently preferred rotting orange in all two-choice experiments except when compared to rotting strawberry (Fig. 1c). Conversely, we observed a very different preference in *Dbus*'s egg-laying choice. Among all the tested

substrates, rotting spinach followed by rotting mushrooms were found to be the most preferred oviposition stimulants and were equally preferred by *Dbus* when compared in a binary choice assay (Fig. 1c and Supplementary Fig. 1d). Additionally, both species exhibited significantly higher attraction to their respective best oviposition substrates in a preference bioassay when tested using big cages (Fig. 1d and Supplementary fig. 1c).

Furthermore, *Dbus* has been reported to share and utilize the same complex ecological niche with five other drosophilids, forming the cosmopolitan guild of *Drosophila*, all of which are known human commensals[21,22]. We hypothesized that the drastic differences in host preference observed might reduce competition and aid in niche separation among co-occurring drosophilids. To test this, we examined oviposition preference between rotting orange and rotting spinach in species pairs, where one species (*Dmel*) remained constant, while the second species varied. In the first three species (*D. simulans*, *D. pseudoobscura*, and *D. hydei*) tested against *Dmel*, we observed significantly higher proportions of eggs laid on plates with the odor of rotting orange (Fig. 1e). In these three pairs we could not morphologically distinguish eggs of individual species. However, almost all eggs, from both species, were found on the side smelling of orange. The pairs containing *D. immigrans* and *Dbus*, where the eggs from the different species could be clearly distinguished, exhibited a gradual shift toward preferring rotting spinach, with *D. immigrans* showing an equal preference for orange and spinach odor and *Dbus* laying almost all eggs on rotting spinach (Fig. 1e).

### Shift in egg-laying behavior in Dbus is mediated by a preference for short-chain oligosulfides

Olfaction plays a crucial role in guiding the egg-laying behavior of drosophilids[32–34], and we sought to identify the key odorants that might be influencing *Dbus*'s oviposition choices. To do this, we focused on the top four oviposition substrates identified (rotting potato, cauliflower, mushroom, and spinach, as shown in Fig. 1c) and analyzed their chemical profiles using SPME-GC-MS, with rotting orange used as a reference. This analysis revealed the presence of 188 different odor molecules (Supplementary Fig. 2a). A principal component analysis (PCA) clearly differentiated the chemical composition of volatiles emitted by rotting oranges from those of the other substrates (Supplementary Fig. 2b).

Further investigation of the chromatograms revealed a significant presence of dimethyldisulfide (DMDS) in rotting mushrooms and rotting spinach, with lower levels detected in rotting cauliflower and rotting potatoes (Fig. 1f). Additionally, dimethyltrisulfide (DMTS) was found in three of the tested substrates, except for mushrooms (Fig. 1f). Based on these observations we hypothesized that DMDS and/or DMTS could be key volatile cues for oviposition site selection in *Dbus*. In the first experiment, the otherwise less attractive odor of orange was supplemented with DMDS or DMTS. After this manipulation, *Dbus* ability to distinguish between rotten orange and rotten spinach was significantly diminished compared to the original preference without any short-chain oligosulfide addition. This indicated the significance of short-chain oligosulfides as crucial oviposition cues for *Dbus* (Fig. 1g).

To assess the role of short-chain oligosulfides alone in stimulating *Dbus*'s oviposition behavior, we conducted a binary oviposition choice assay. In this experiment, the flies showed a significant preference for agarose plates perfumed with DMDS (Fig. 1h) over the control containing mineral oil. However, when presented with DMTS alone, the flies did not exhibit a significant oviposition preference (Fig. 1h). Intriguingly, a 1:1 ratio of DMDS and DMTS was highly preferred (Fig. 1h). However, the oviposition index of the binary blend was not significantly different, and without a synergistic effect, when compared with the individual oviposition indices of either DMDS or DMTS

 

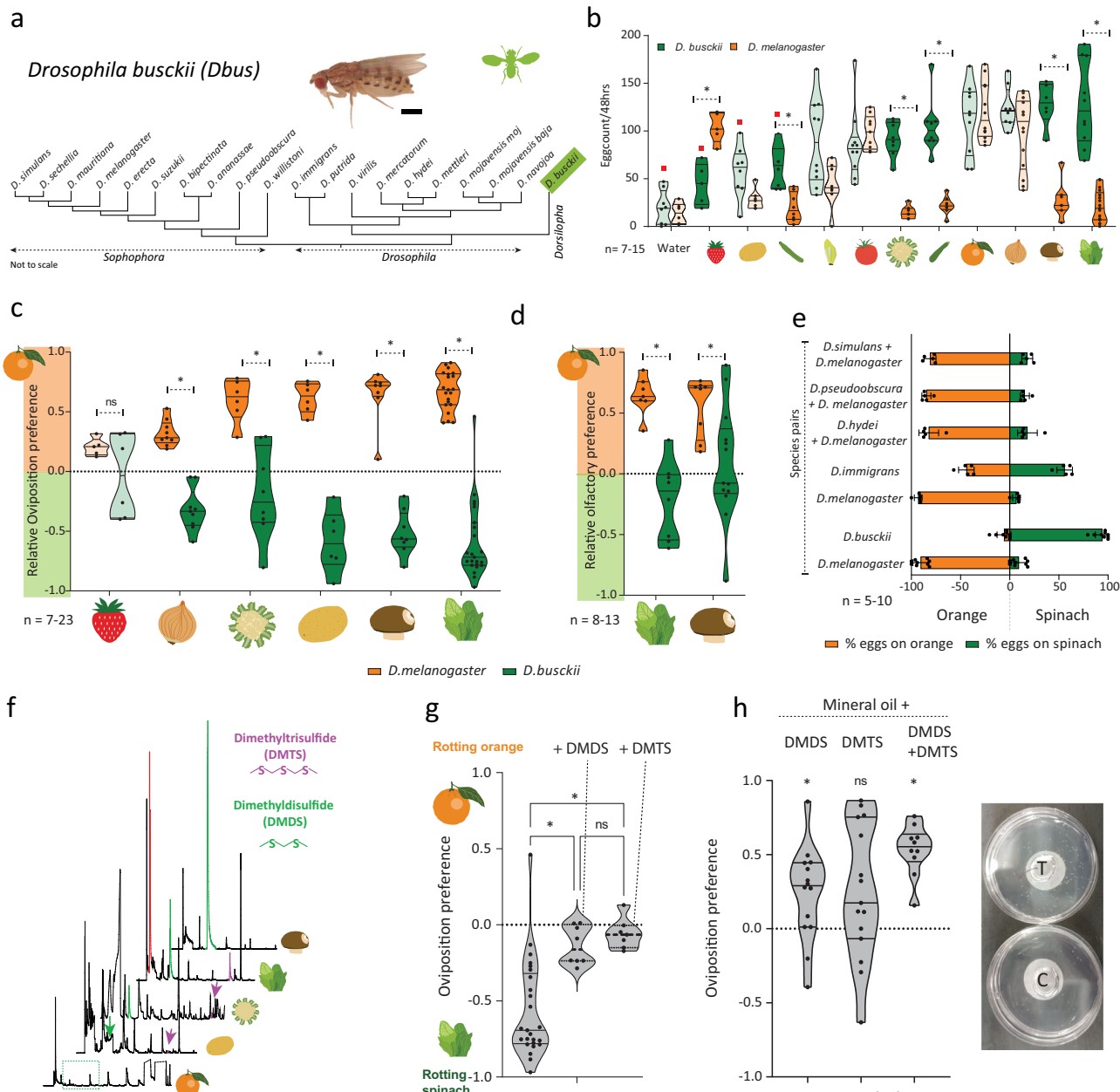

(p = 0.6442). These experiments demonstrated the importance of short-chain oligosulfides in guiding *Dbus*'s egg-laying choices. Furthermore, despite rotting spinach also emitting substantial amounts of dimethylsulfide (DMS), this compound did not elicit oviposition (Supplementary Fig. 1e). Importantly, our investigations excluded the influence of feeding stimulants such as yeast powder or sucrose in the oviposition plates, confirming that the observed oviposition choices were predominantly driven by olfactory cues.

Lastly, we examined whether DMDS triggered oviposition in *Dmel* and found that *Dmel* laid significantly fewer eggs in the presence of DMDS compared to *Dbus* (Supplementary Fig. 1f). The calculated oviposition index was not significantly different from the control (Supplementary Fig. 1f). Moreover, *Dmel*'s egg-laying activity was notably low within the 48-h trial period, and fly mortality increased towards the end of the trials (data not quantified but see below). In summary, our findings highlight the critical role of short-chain oligosulfides, particularly DMDS, as key oviposition cues guiding the egg-laying behavior of *Dbus*.

## Dbus features dedicated olfactory sensory neurons to detect DMDS

The significant role of DMDS as an oviposition cue in *Dbus* led us to search for olfactory sensory neurons (OSNs) responsible for detecting short-chain oligosulfides. To gain a comprehensive understanding of the peripheral olfactory system, we conducted single sensillum recordings (SSR) from all basiconic sensilla located on the antennal third segment of *Dbus*. We used a panel of 43 chemically diverse compounds, known to be ecologically relevant for various *Drosophila* species, as stimuli (Supplementary Table 1)[7,12,13]. For comparison, we also tested the ten well-known antennal basiconic sensillum classes in *Dmel* with the same odor spectrum and dilutions. In our recordings from *Dbus*, we identified eleven basiconic sensillum classes, out of which four were comparable to four *Dmel* sensillum types (*Dmel*ab1, ab4, ab6, and ab9[12,13,35]). However, we also encountered seven classes in *Dbus* where a clear homology to the known sensillum types in *Dmel* could not be directly established (Supplementary Fig. 3a, b). To avoid confusion, we assigned the prefix "B" to all sensillum classes in *Dbus*,

**Fig. 1 | Host shift in *D. busckii* is mediated by preference for short-chain oligosulfides, specifically DMDS. a** A female *D. busckii* (*Dbus*). Phylogenetic relationship between species from three subgenera within the family Drosophilidae. Branch lengths are representative and not to scale. **b** No-choice bioassay experimental setup and number of eggs laid by each species during 48 h. Significance tested between egg counts of each species. Darkened violin plots indicate significant differences between the number of eggs laid by each species (two-tailed, unpaired t-test with Welch's correction. Significance p < 0.05). Furthermore, significance was also tested between the number of eggs laid by *Dbus* against several substrates (one-way ANOVA with multiple comparisons). A filled red box indicates no significant difference in egg numbers when presented with corresponding substrates and compared to water. *Dbus* and *Dmel* are represented by green and orange colors, respectively, in either darkened or light color palettes. Note that all substrates were used in a rotting state. **c** Binary choice assay testing relative oviposition preference (ROP) between rotting orange and a second rotting substrate. Darkened violin plots indicate significant differences between oviposition indices tested between *Dmel* and *Dbus* (two-tailed, unpaired t-test with Welch's correction. Significance p < 0.05). **d** Binary choice assay testing attraction between rotting oranges and spinach/ mushroom in a BugDorm cage arena (see "Methods" section). Significance was tested using two-tailed t-test with Welch's correction. *p < 0.05. (p < 0.0001 and p = 0.0069 respectively). **e** Binary choice experiment to test niche separation mediated by a preference for two substrates between two species, where one always was *Dmel*. The first three rows depict a combined

measure of the percent of eggs laid by both species as the eggs could not be morphologically differentiated from each other. The bottom two rows depict the percent eggs laid on each substrate by individual species as it was possible to visually differentiate species-specific eggs. n = 5–9. Error bars represent mean ± SD. **f** SPME-GC-MS chromatograms of four rotting substrates on a normalized abundance scale. Peaks representing dimethyldisulfide (DMDS) and dimethyltrisulfide (DMTS) are highlighted in green and magenta, respectively. Dimethylsulfide (DMS) was found only in rotting spinach and is highlighted in red. **g** Binary choice between rotting spinach vs rotting oranges perfumed with DMDS or DMTS (10 μl, $10^{-2}$ in mineral oil each) when tested with *Dbus*. Significance was tested between control (only rotting orange choice) and treatments using one-way ANOVA followed by multiple comparisons and testing significance between control (column one, only rotting orange choice) and treatments (rotting orange + DMDS/DMTS). ns: p > 0.05, *p < 0.05. The control data (spinach vs orange) was replotted from (**c**) and reused to compare with the additive effect of short-chain oligosulfides. (column 1 vs 2, p = 0.003; column 2 vs 3, p > 0.9; column 1 vs 3, p = 0.0005). **h** Binary choice assay testing oviposition preference between DMDS (10 μl, $10^{-2}$ in mineral oil) and mineral oil control in a BugDorm cage arena (see "Methods" section) when tested with *Dbus*. Significance was tested using two two-tailed, unpaired t-test with Welch's correction. ns: p > 0.05, *p < 0.05. (DMDS vs control: p = 0.014; DMTS vs control: p = 0.0538; DMDS + DMTS vs control: p < 0.0001) Source data are provided as a source data file.

independently from the names assigned in other species (Supplementary Table 2).

Subsequently, we screened the eleven basiconic classes in *Dbus* and the ten classes in *Dmel* with DMDS and identified only one sensillum class (in *Dbus*) responding to DMDS even at low concentrations ($10^{-4}$ v/v). This sensillum was named Bab2 and displayed spontaneous activity from two OSNs, distinguishable based on action potential amplitudes (Fig. 2a). The Bab2A OSN (with larger action potentials, Supplementary Fig. 3d) responded to low molecular weight compounds such as acetone or 2-butanone, while the Bab2B OSN (with smaller action potentials) exhibited narrow tuning to short-chain oligosulfides (Fig. 2b). When comparing sensitivity, we found that the Bab2B OSN was most responsive to DMDS, followed by DMTS, and least responsive to dimethylsulfide (DMS) and dipropyldisulfide (DPDS) (Fig. 2c, d and Supplementary Fig. 3c). This demonstrated that in comparison to *Dmel*, *Dbus* possesses a specific OSN type, with high specificity and sensitivity to oligosulfides, particularly to DMDS (Fig. 2e).

To understand whether the detection of DMDS is a gradual gain or loss of response across the *Drosophila* phylogeny, we performed SSRs from sensilla on the posterior-proximal region of the third antennal segment (i.e., where Bab2 is located in *Dbus*) in ten distantly related drosophilids. We observed that Bab2A OSNs respond to acetone & 2-butanone (Supplementary Fig. 3a). It has been reported that the receptor expressed in ab2A OSNs (Or59b) in *Dmel* responds to both of these abovementioned compounds as well as to methyl acetate. Moreover, the neighboring OSN type (*Dmel* ab2B expressing Or85a) responds to ethyl-3-hydroxybutyrate and isopropyl benzoate[12,36]. Based on this knowledge, next, we hypothesized that a conserved sensillum type potentially housing an OSN responding to DMDS would display an A neuron responding to methyl acetate and/ or 2-butanone, while the neighboring B neuron would respond to ethyl-3-hydroxybutyrate and/or isopropyl benzoate and hypothetically also to DMDS (Fig. 2f and Supplementary Fig. 4a). We could indeed identify such a conserved sensillum class in all species investigated, responding to key diagnostic ligands at varying strengths (Fig. 2f). We then challenged these sensilla with DMDS and could observe a pattern of gradual gain of response to DMDS when transitioning from *Dmel* to *Dbus* (Fig. 2f, g). In summary, our findings demonstrate the presence of an OSN class in *Dbus* that is narrowly tuned to DMDS.

## Dbus has evolved tolerance against DMDS

Our investigations so far revealed that *Dbus* displays a preference for ovipositing on substrates emitting DMDS, which raises the possibility that the species is frequently exposed to this compound. However, DMDS is widely used as a fumigant and has neurotoxic properties[28,29]. This intriguing contradiction prompted us to conduct survival experiments with *Dbus* and five other species forming the cosmopolitan guild (as tested in Fig. 1e) when exposed to food mixed with DMDS ($10^{-3}$ v/v). These experiments demonstrated an exceptional survival ability of *Dbus* on food containing such relatively high levels of DMDS, whereas all the other species tested showed significantly higher and rapid mortality within four hours (Supplementary Fig. 5a). We also observed an intermediate tolerance phenotype in *D. ananassae* and *D. hydei*, as these flies exhibited a delayed susceptibility pattern.

It is known that DMDS exerts its neurotoxic effect by noncompetitively binding to the mitochondrial cytochrome c oxidase (COX), also known as complex IV, leading to the inhibition of ATP generation[28]. This, in turn, triggers the activation of the ATP-dependent potassium channel (K-ATP), causing cellular hyperpolarization[28] (Fig. 3a).

During our experiments, flies were kept and exposed to DMDS mixed with food (Supplementary Fig. 5b1). Hence, we first sought to establish in which phase (respiratory or via ingestion) DMDS acted on the flies. We performed a round of experiments (Supplementary Fig. 5b2), where only DMDS vapors were presented to the test flies[37]. We observed a knock-down effect similar to the one observed in our experiments involving DMDS mixed with fly food. This strongly indicated the involvement of the respiratory pathway in DMDS susceptibility. Notably, the knock-down effect was temporary and reversible in *Dmel* up to five hours post-exposure (Supplementary Fig. 5c). Such a reversible knock-down effect has been previously reported with other toxic compounds, such as cyanide, that function by targeting mitochondria and hindering cellular respiration[38,39]. Therefore, such a reversible effect of DMDS in *Dmel* indicated toward the involvement of mitochondria in DMDS susceptibility. Furthermore, exposure to known insect anesthetics, such as sevoflurane[37], showed no differences in anesthesia tolerance between *Dbus* and *Dmel* (Supplementary Fig. 5e).

Next, we tested the tolerance of *Dbus* and *Dmel* to DMDS at varying concentrations and found that DMDS susceptibility was dose-dependent (Fig. 3b and Supplementary Fig. 5d). We identified a critical concentration ($3 \times 10^{-3}$ v/v) that showed a significant difference in

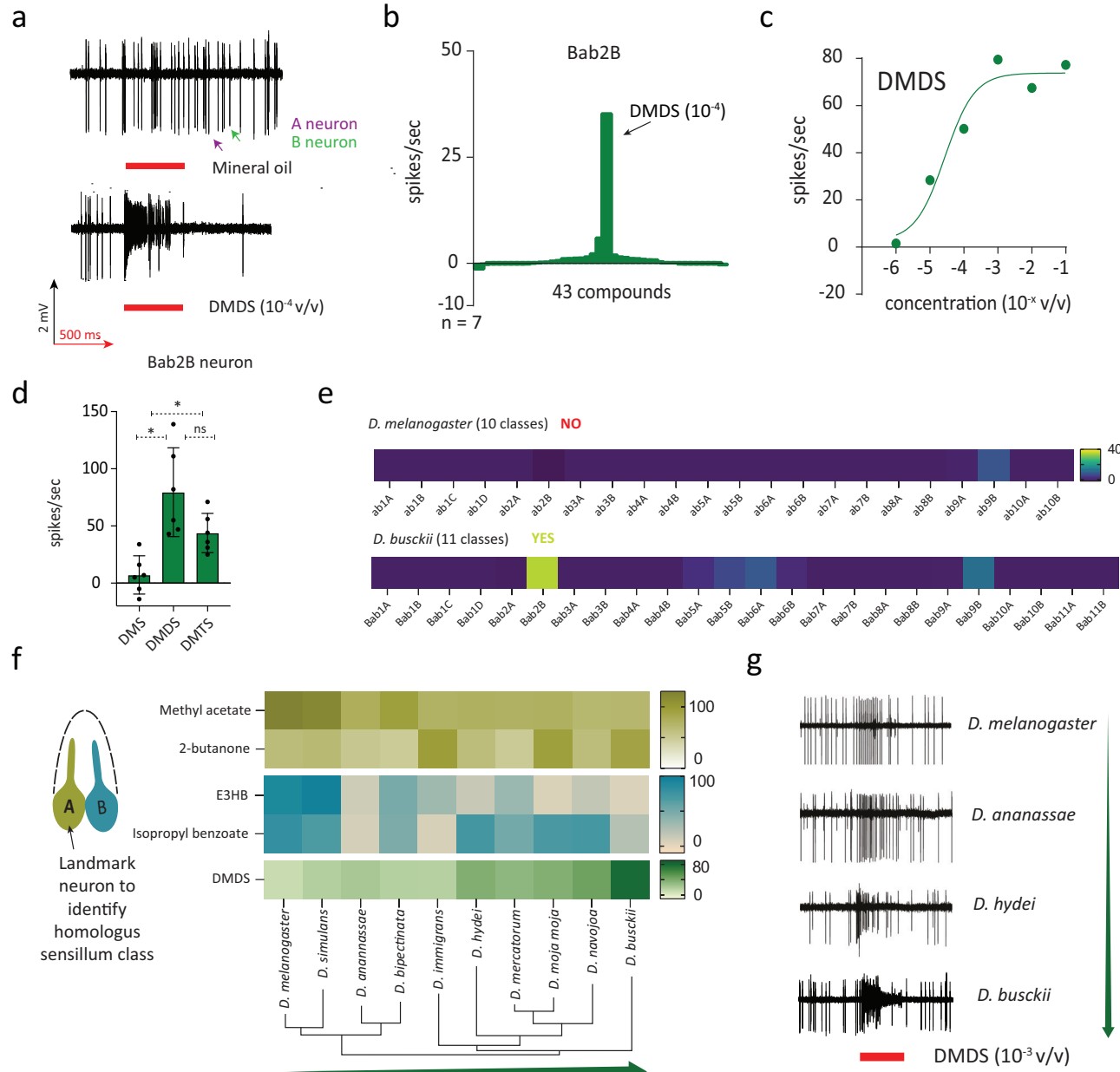

**Fig. 2 | Screening of *D. busckii* antenna reveals 11 basiconic sensillum classes with Bab2B OSN type narrowly tuned to oligosulfides. a** Representative traces of an extracellular recording from the Bab2 sensillum class. Responses to mineral oil (control) and DMDS ($10^{-4}$ in mineral oil) are shown. **b** Tuning width of the Bab2B OSN class. The OSN type is narrowly tuned to DMDS. **c** The dose–response properties of the Bab2B OSN indicate high sensitivity to DMDS. n = 5. **d** Responses of the Bab2B OSN type to stimulation with $10^{-3}$ v/v of linear, short-chain oligosulfides. n = 6. Each replicate corresponds to a single Bab2B OSN recording from an individual fly. (one-way ANOVA with multiple comparisons, *p < 0.05, ns p > 0.05). Error bars represent mean ± SD. **e** Screening of all known basiconic types in *Dmel* and *Dbus* with DMDS ($10^{-4}$ in mineral oil) revealed DMDS detection predominantly by the Bab2B OSN class in *Dbus*. A comparable response was not observed at this concentration from *D. melanogaster* OSNs. **f** Heatmap representation of SSR data (represented as spikes/s) for recordings from Bab2-like sensilla in the posterior-proximal region of the antenna of multiple *Drosophila* species. n = 3–5. **g** Representative SSR traces of homologous OSNs responding to DMDS ($10^{-3}$ v/v) when tested across multiple *Drosophila* species. Source data are provided as a source data file.

susceptibility between the two species, which was then used for subsequent experiments (Fig. 3b). To investigate the involvement of COX in the DMDS tolerance phenotype, we used sodium azide ($NaN_3$), known to be an exclusive COX inhibitor along with cyanides and carbon monoxide[40,41]. Testing $NaN_3$ in a dose-dependent manner revealed a key concentration at which a similar tolerance phenotypic difference, comparable with DMDS, was observed. This result hinted towards clearly differential COX functional kinetics between the two species (Fig. 3c and Supplementary Fig. 5d). COX is a multimeric protein complex comprising 14 subunits, in which three catalytic subunits

(COX I-III) are encoded by the mitochondria, while the remaining 11 subunits are of nuclear DNA origin and serve a structural role (Fig. 3d). Protein sequence comparisons of COX I subunits among five species (tested in Supplementary Fig. 5a) revealed differences in two amino acid positions (aa108 & aa331) that could potentially correlate to the observed DMDS tolerance phenotypes. As a result, we hypothesized that amino acids at these two positions would be pivotal in determining the DMDS tolerance phenotypes (Fig. 3d).

To test our hypothesis, we compared COX I protein sequences across the Drosophilidae family, including sequences from

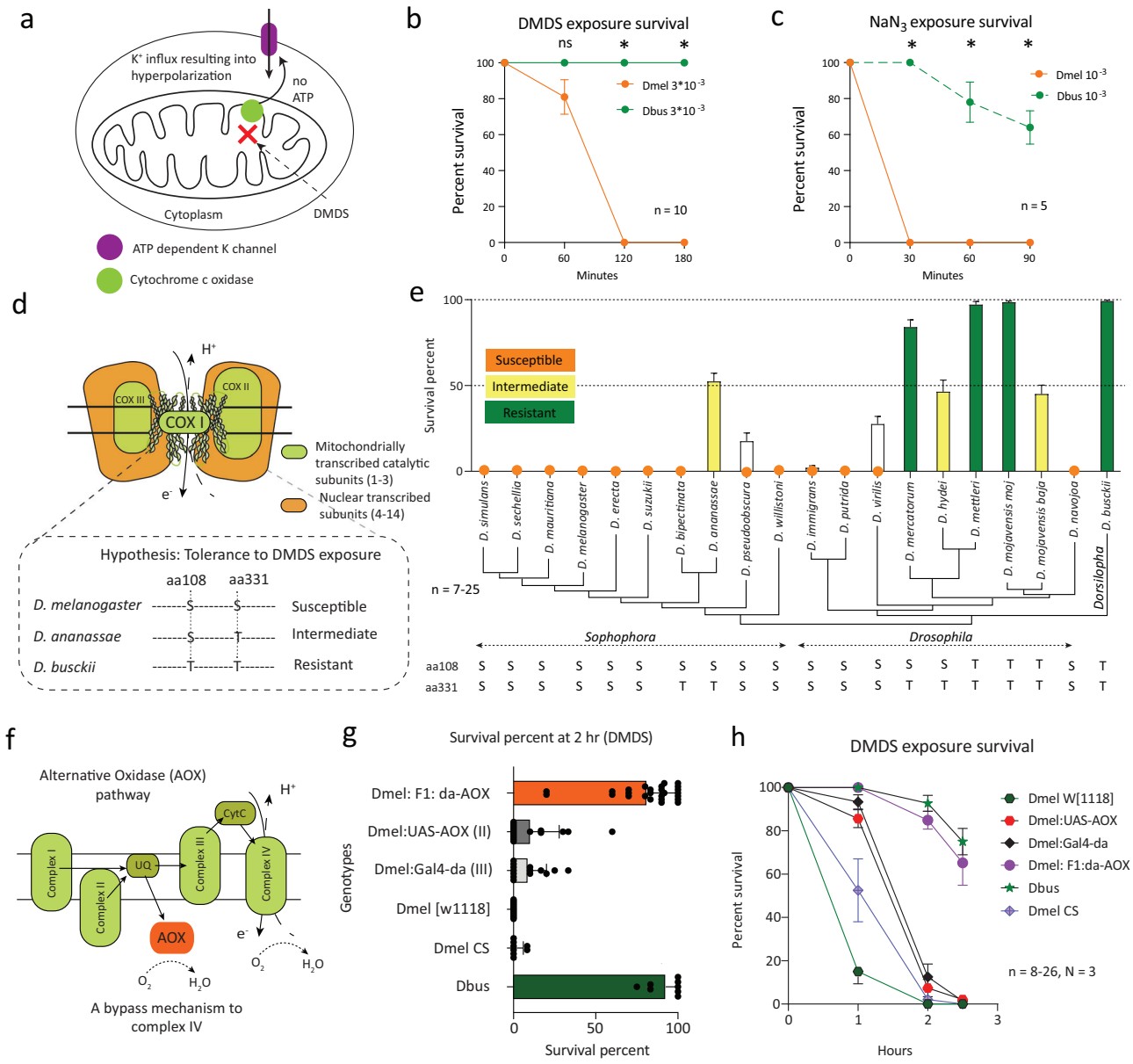

**Fig. 3 | D. busckii has evolved to tolerate high concentrations of DMDS.**
**a** Schematic representation of one of the mechanisms involved in DMDS-induced inhibition. Adapted from Dugravot et al.[28]. **b** Time course experiment demonstrating differences in tolerance to DMDS, when compared between *Dbus* and *Dmel*, at the given concentration. Mann–Whitney test or one sample t-test for data set with identical values: *p < 0.05, ns p > 0.05. Error bars represent mean ± SEM. (for DMDS exposure, t = 60 min, p = 0.0791; for NaN₃ exposure, t = 60 min, p = 0.0022). **c** Time course experiment demonstrating differences in tolerance to NaN₃, when compared between *Dbus* and *Dmel*, at the given concentration. Mann–Whitney test: *p < 0.05, ns p > 0.05. Error bars represent mean ± SEM. **d** Schematic representation of the cytochrome c oxidase (COX) protein made up of 14 subunits. Box enclosed by a dotted line represents the hypothesis behind the involvement of two key amino acids (positions aa108 & aa331) conferring different degrees of tolerance to DMDS. **e** Tolerance experiments demonstrating variation in DMDS tolerance in multiple *Drosophila* species tested across the phylogeny. The

y-axis depicts survival percent at t = 4 h post-DMDS exposure. Different color codes are used to represent tolerance categories. Species showing survival levels below 50% were categorized as susceptible, those within a range of 50–75% were intermediate, while survival above 75% was considered completely tolerant. Amino acids hypothesized to be involved in conferring tolerance are represented beneath each species. n = 7–25. Error bars represent mean ± SEM. **f** Schematic representation of the AOX pathway showing upstream positioning of AOX providing a bypass electron transfer route. **g** Survival percentage of flies at t = 2 h. post-DMDS exposure using an experimental setup as described in Supplementary Fig. 5b2. The x-axis represents the percentage of flies alive after exposure to DMDS for 2 h. Whereas the y-axis denotes multiple genotypes tested in the study. n = 8–26. Error bars represent mean ± SD. **h** A time course representation of DMDS-induced susceptibility across *Dbus* and multiple genotypes in *Dmel* including F1 *Dmel* progeny expressing AOX under the control of *daughterless* promoter explained in (**g**). n = 8–26. Error bars represent mean ± SEM. Source data are provided as a source data file.

approximately 200 species available from the NCBI server. This comparative analysis revealed six other drosophilid species with plausible DMDS tolerance (i.e., sequence similarity with *Dbus* at both crucial amino acid positions) and seven additional species with possible intermediate tolerance (i.e., sequence similarity with *Dbus* at position

aa108). We exposed multiple *Drosophila* species (selected from the sequence comparisons) to DMDS, and our prediction held true for 17 out of 20 species tested (an 85% success rate) (Fig. 3e and Supplementary Fig. 5f). We also found *D. mojavensis mojavensis* as another species with high DMDS tolerance (Fig. 3e) and possessing the same

two crucial amino acids as *Dbus*. However, this species is highly unlikely to encounter DMDS in its natural niche, and showed a neutral egg-laying preference when tested against DMDS (Supplementary Fig. 5g). Importantly, there was no correlation between the reported global origin of a given species and the observed DMDS tolerance phenotype from our results (Supplementary Fig. 5i). Thus, our findings fit our prediction regarding the crucial amino acids, but future direct evidence should be obtained with appropriate mitochondrial gene editing.

Next, we aimed to &demonstrate the involvement of COX in governing these tolerance phenotypes. Many animals exposed to extreme environments express an alternative oxidase (AOX), which functions similarly to COX but is reportedly resistant to known COX inhibitors[42–44]. AOX is located upstream of COX in the mitochondrial electron transport chain and serves as a bypass mechanism if COX is inhibited (Fig. 3f). We hypothesized that expression of AOX (upstream of COX) in a DMDS susceptible species such as *Dmel* would provide a bypass mechanism and confer tolerance to DMDS at least to some extent. To test this hypothesis, we expressed AOX under the regulatory control of the *daughterless* (*da*) gene. Our results revealed that AOX expression in *Dmel* successfully conferred tolerance to DMDS compared to parental and species genotype controls (Fig. 3g, h). This clearly indicates that COX is indeed involved in DMDS tolerance and a redundant mechanism is sufficient to rescue the respiratory chain from DMDS susceptibility.

## DMDS tolerance in Dbus adults is reflected also in larvae

We discovered that adult *Dbus* flies can tolerate the toxic compound DMDS. Similarly, we also investigated the attraction of *Dbus* larvae to DMDS. A clear attraction was noted (Fig. 4a), further supporting the role of DMDS as a cue for suitable food sources for larvae and, thereby, for beneficial oviposition sites. We also checked the effect of DMDS exposure ($10^{-2}$ v/v conc.) on the activity of *Dmel* larvae. However, we observed significantly less activity in the case of *Dmel* larvae. Furthermore, it is likely that *Dbus* larvae are exposed to even higher DMDS concentrations as compared to adults. Therefore, we investigated whether *Dbus* and *Dmel* larvae could complete their life cycle on DMDS-emitting substrates, such as rotting spinach and mushrooms, as

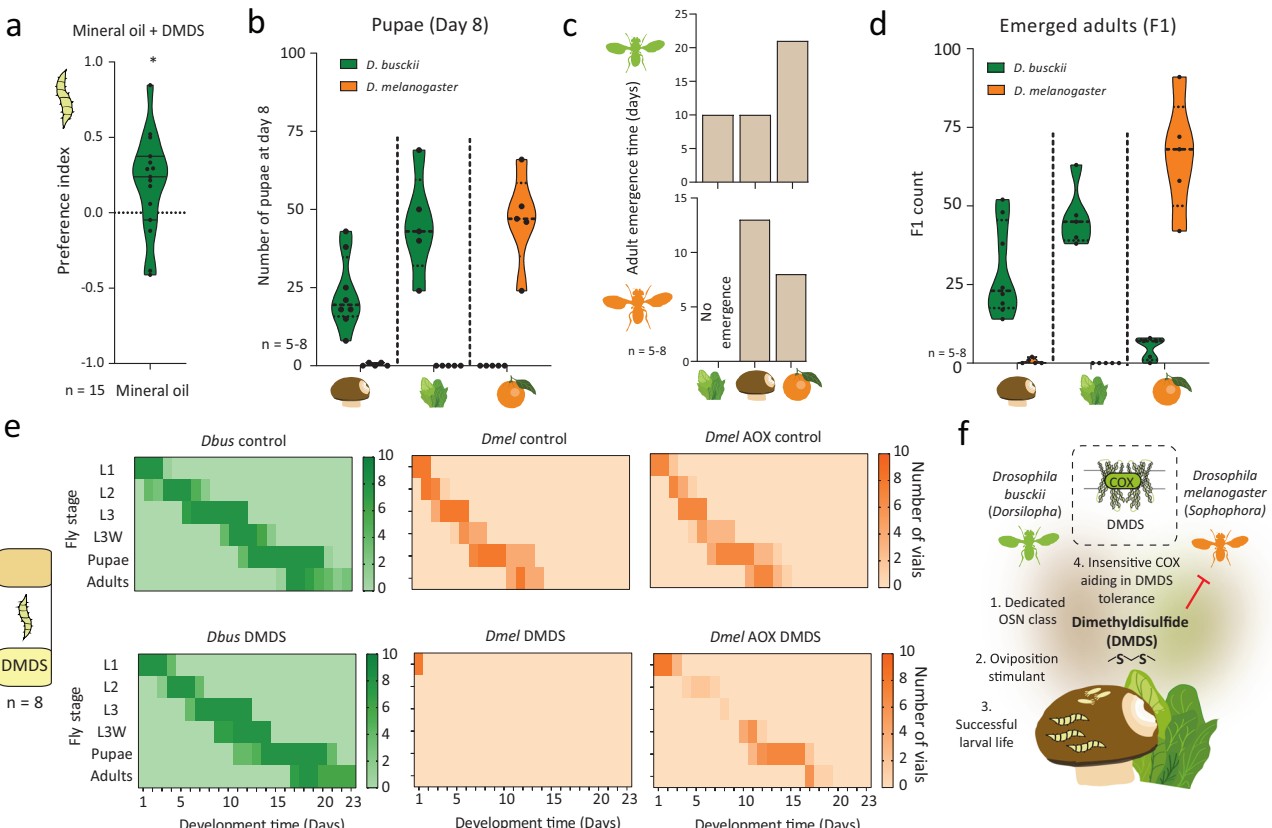

**Fig. 4 | DMDS-emitting substrates are sufficient for the life cycle completion of *D. busckii* and *D. busckii* larvae show complete tolerance to DMDS. a** *D. busckii* larval choice assay to test preference between DMDS (10 µl, $10^{-2}$ in mineral oil) and mineral oil control in a Petri plate (Supplementary fig. 1c, see "Methods" section). Significance was tested using two-tailed, unpaired t-test with Welch's correction. *p = 0.0436. **b** Number of pupae recorded from the vials on day 8 post parental adults' transfer. Significance was tested between species and independently for each substrate using two-tailed, unpaired t-test with Welch's correction. *p < 0.05. (from left to right and tested between *Dbus* and *Dmel*; p = 0.0009, p = 0.0035, p = 0.0023 respectively). **c** Number of days required for adult emergence for either species when subjected to different substrates. *Dbus* developed significantly faster on either mushrooms or spinach as compared to oranges, while the converse was true for *Dmel*. **d** Number of F1 adults that emerged from each substrate. Filled plots show significant differences when compared between *Dbus* and *Dmel*. Note that the number of pupae for *Dbus* or *Dmel* is zero on day 8 indicating slower growth (i.e.,

the longer time required to pupate) and hampered nutrition on unfavorable substrates. Significance was tested between species and independently for each substrate using a two-tailed, unpaired t-test with Welch's correction *p < 0.05. (from left to right and tested between *Dbus* and *Dmel*; p = 0.0009, p = 0.0005, p = 0.0013 respectively). **e** Developmental period comparison within species when 50 L1 larvae were kept on food containing DMDS ($10^{-3}$). No difference between larvae exposed to DMDS compared to control conditions was observed in the case of *Dbus* (left panel), while *Dmel* larvae were highly susceptible to DMDS and died within a couple of hours post-exposure (middle panel). Another group of larvae, where ~50% should contain AOX, showed partial tolerance and a few individuals developed successfully until the adult stage (right panel). n = 8 where a single replicate is defined by a single vial. **f** A schematic overview for *Dbus* host choice representing multiple evolutionary adaptations revolving around DMDS. Source data are provided as a source data file.

compared to the control substrate, fermenting orange. We found that *Dbus* larvae were able to successfully complete their life cycle on either spinach or mushrooms but performed poorly on oranges (Fig. 4b–d). Conversely, *Dmel* larvae reached the pupal stage only on oranges and developed extremely poorly on mushrooms (Fig. 4b–d). To further understand the developmental dynamics of the larvae, we conducted experiments using food supplemented with synthetic DMDS to understand development dynamics of larvae (Fig. 4e). *Dbus* larvae were unaffected by the presence of DMDS in the food, and showed no difference compared to the controls without DMDS (Fig. 4e). However, *Dmel* larvae were highly susceptible to DMDS, with larval mortality observed within a few hours of exposure to the food (Fig. 4e). As a final test, we introduced *Dmel* larvae expressing AOX under the control of the *daughterless* (*da*) promoter. As expected, bypassing the COX complex allowed larvae to survive in this medium and, in some cases, adults to be produced (Fig. 4e).

In conclusion, our study unveils an evolutionary strategy employed by a generalist, cosmopolitan drosophilid that demonstrates both ovipositional preference for and tolerance to DMDS, a compound toxic to other tested drosophilids (Fig. 4f). This unique trait enables *Dbus* to thrive on substrates rich in DMDS, setting it apart from its counterparts and highlighting its exceptional adaptation to toxic environments.

## Discussion

Among drosophilid flies, *Dbus* stands out due to several distinctive features. These include its unique phylogenetic position and a notable preference for breeding in rotting vegetables rather than fermenting fruits, a characteristic that distinguishes it from other species within the *Drosophila* genus[20,45]. These distinctive traits make *Dbus* an intriguing subject for studying the impact of shifted selection pressures in the *Drosophila* genus. Our research reveals that *Dbus* exhibits a specific preference for unusual oviposition substrates, which is influenced by a behavioral inclination towards short-chain oligosulfides, particularly dimethyldisulfide (DMDS). Through further investigation, we identify a dedicated type of olfactory sensory neuron (OSN) located in the basiconic sensilla on the *Dbus* antenna, specifically attuned to DMDS. Additionally, we establish that *Dbus* is capable of completing its life cycle on substrates known to release DMDS, as well as on artificial substrates containing synthetic DMDS. Notably, our findings contrast with the known toxic effects of DMDS on many other insects, even to the extent of its use as a fumigant[29].

Remarkably, our research reveals that, unlike several other *Drosophila* species we tested, *Dbus* has evolved a tolerance and survival mechanism in the presence of DMDS. In a series of experiments, we demonstrate that this DMDS tolerance in *Dbus* can be attributed to the insensitivity of its mitochondrial cytochrome oxidase (COX), a known target site for DMDS-mediated inhibition in other insects[28].

The existing literature pertaining to the natural ecology of *Dbus* is notably limited. While it suggests an origin for *Dbus* in the tropical forests of southeastern Asia[30], there is a conspicuous absence of information regarding its possible ancestral diet. However, field collections of *Dbus* provide substantial evidence to consider this species as exhibiting a generalist feeding and breeding behavior biased towards vegetables and fungi[20,24,25,27]. Our findings align with previous studies reporting the collection of *Dbus* from various vegetable substrates, including rotting cauliflower and potatoes[20]. Importantly, we demonstrate that many of these substrates release short-chain oligosulfides during fermentation, as illustrated in Fig. 1g. To the best of our knowledge, this marks the first report highlighting the ecological significance of such short-chain oligosulfides for any *Drosophila* species.

The use of short-chain oligosulfides as semiochemicals by insects has been documented in other contexts, involving mosquitoes, bed bugs, blow flies, parasitic wasps, cabbage root flies, and carrion-mimicking flower breeding flies[46–50]. Decaying carrion and carrion-mimicking flowers have been found to emit significant amounts of both DMDS and DMTS in their bouquets[47,50,51]. However, non-toxic DMTS appears to be the dominant volatile associated with these niches[50].

Our research revealed the emission of both of these aforementioned short-chain oligosulfides from substrates preferred by *Dbus*. Nevertheless, when tested individually, DMTS alone did not induce a significant oviposition preference compared to DMDS. This discovery is intriguing, as multiple studies have demonstrated the use and prevalence of DMTS over DMDS for navigation and oviposition in other species, such as carrion-eating flies like *Lucilia sericata* and *Calliphora vicina*. Furthermore, electroantennogram studies indicated strong responses to DMTS but limited to no response to DMDS[51,52]. Given that there are no reports of *Dbus* being captured from carrion or carrion-mimicking flowers, it is reasonable to hypothesize that this shift in ligand preference between these two structurally related oligosulfides could be a contributing factor in the discrimination of food and oviposition sites.

Additionally, mycophagy in drosophilids, including *Dbus* and *D. falleni*, has been well-documented[53,54]. However, the specific odors that mediate the attraction of drosophilids to mushrooms remain largely unknown. Our results demonstrate that DMDS is one of the compounds involved in mediating the preference of *Dbus* for oviposition on mushrooms. This observation is consistent with prior studies reporting the emission of sulfur compounds, including DMDS, from mushrooms[55,56].

In parallel, our single sensillum recordings from neurons on the *Dbus* antenna allowed us to characterize an olfactory sensory neuron (OSN) type that responds specifically to DMDS. Within the genus *Drosophila*, OSNs responding to DMDS have previously been reported only in *D. mojavensis* and *D. novamexicana*, which specialize in fermenting cacti and fermenting slime flux, respectively[6,7]. However, the ecological significance of DMDS for these species remains unclear, as DMDS is not present in the odor bouquets of their known natural hosts[57].

In our study, we specifically screened OSNs present in basiconic sensilla in *Dbus*, as these neurons are generally known to detect food odors[13]. In contrast, OSNs in trichoid sensilla are associated with pheromone detection, while neurons present in coeloconic sensilla are involved in detecting acids and amines. Furthermore, the OSN type identified in *Dbus* (Bab2B) exhibited high specificity in responding to DMDS, even at low concentrations (as low as $10^{-6}$ v/v), indicating its likely role in the primary circuit for DMDS detection.

Moreover, we detected the presence of DMDS in various substrates originating from different plant families, including *Brassicaceae*, *Amaranthaceae*, and *Solanaceae*. This suggests that the origin of DMDS is likely independent of the specific plant family. DMDS is a well-established bacterial biomarker and plays a critical role in the natural sulfur nutrition cycle[58–60]. Additionally, it serves as a distinctive volatile compound associated with plants in the *Brassicaceae* family[60,61].

Considering the well-established role of DMDS as a bacterial biomarker, it is reasonable to hypothesize that bacteria associated with potato rot or mushroom rot may be involved in attracting and subsequently being transferred by *Dbus*, similar to how yeast volatiles attract *Dmel* aiding in the transport of yeasts from one site to another[34]. This possibility gains support from existing reports indicating that *Dbus* is a significant commercial pest and vector of bacteria, such as *Erwinia sp.*, which are responsible for causing soft rot diseases[27]. We acknowledge that DMDS-based host preference may represent just one of several ecologically relevant factors influencing oviposition in *Dbus*. Furthermore, it is important to recognize that the host preference we observed in our experiments is contingent upon the rotting stage, and variations in host choice may occur depending on the relative stage of decay between paired substrates. Lastly, while successful oviposition may not always guarantee

subsequent development, our no-choice bioassay revealed a substantial number of *Dbus* eggs on oranges. However, when we assessed the completion of the life cycle, we discovered a significantly hindered development in fermenting oranges, where *Dbus* larvae appeared to become arrested in the first larval instar stage. This observation aligns with the very first report of *Dbus* in 1911, where a large number of *Dbus* eggs were observed on fruits without any subsequent adult emergence[62].

We observed an exceptional level of tolerance in *Dbus* to high concentrations of DMDS. It is worth noting that this tolerance, while remarkable, still displayed dose-dependent characteristics, as lethal concentrations of DMDS could be reached for *Dbus* as well. DMDS exerts its effects in a manner akin to cyanide and azides, binding to cytochrome c oxidase (COX)[28,39,40]. The mode of action of cyanide toxicity has also been primarily attributed to a non-linear binding to COX, displaying dose-dependent kinetics in inhibiting cellular respiration[39]. Our findings concerning DMDS align with reported kinetics, indirectly supporting COX inhibition as a major consequence of DMDS exposure in the tested flies. Furthermore, we found that exposure to $NaN_3$, a known COX-specific inhibitor[63], resulted in a similar dose-dependent tolerance in *Dbus*, which exhibited approximately tenfold greater tolerance to $NaN_3$ compared to *Dmel*.

To delve deeper into this, we demonstrated that by providing a redundant electron transfer mechanism, which effectively bypasses the COX-mediated transfer channel, we could rescue *Dmel* from susceptibility to DMDS. This rescue strategy involved transiently expressing an alternative oxidase (AOX) upstream of COX. Our results are consistent with previous studies that have reported rescue from cyanide toxicity through the transient expression of AOX in *Dmel* and human cells[44,64]. Unlike COX, AOX is encoded by nuclear DNA. To address how AOX gains access to the mitochondria, it is noteworthy that although not extensively explored, prior reports suggest an import of AOX through the mitochondrial membrane, followed by integration into the electron transfer chain[64].

Additionally, another mechanism involving the activation of $Ca^{2+}$-dependent potassium channels by DMDS has been documented[65]. However, we argue that since these channels belong to a large family of potassium channels and could have multiple redundant proteins acting as DMDS targets, it would be challenging to pinpoint a single target[66]. Furthermore, as our experiments with AOX were sufficient to rescue DMDS susceptibility in *Dmel*, COX appears to be the primary target. In summary, our findings strongly suggest the involvement of *Dbus* COX as a prominent target site, if not the sole mechanism, for DMDS tolerance.

Previous studies have postulated that DMDS tolerance could be due to the presence of insensitive proteins, as opposed to other detoxification mechanisms. For instance, when considering detoxification mechanisms in *Allium* specialist insects, such as *Acrolepiopsis assectella*, exposure to DMDS did not alter the levels of glutathione-S-transferase (GST), suggesting that tolerance could be due to an insensitive target site[67].

Moreover, our results demonstrate the exceptional survival of *Dbus* larvae on food containing synthetic DMDS. Given that insect larvae spend a significant portion of their developmental phase within the food source, *Dbus* larvae are likely exposed to DMDS for supplementary periods. While we cannot rule out the possibility of other detoxification mechanisms, particularly those related to feeding-based detoxification, playing a role in DMDS tolerance, our findings involving *Dmel* larvae expressing AOX (Fig. 4e) point towards the involvement of mitochondria as a factor conferring DMDS resistance during larval stages.

When exposing adult *Dmel* to natural substrates such as rotting spinach or mushrooms, we observed no adverse effects (data not quantified). However, the most significant implications were observed

in terms of substantially impaired larval development and F1 emergence (Fig. 4b, d). Therefore, based on our results, it could be hypothesized that tolerance to DMDS may be a strategy of particular importance for *Dbus* larvae due to their frequent and very close proximity to DMDS-emitting substrates. This tolerance mechanism may be retained passively during the adult stages.

Finally, *Drosophila* species are under constant threat of being attacked by parasitic wasps or nematodes[68–71]. Volatiles associated with the preferred oviposition sites have been demonstrated to be involved in conferring protection by repelling parasitic wasps[32]. Similarly, it would be interesting to test if DMDS confers protection to *Dbus* larvae and has a defensive potential against parasitic wasps or nematodes.

We propose the involvement of two specific amino acids that may potentially desensitize the target COX protein to the inhibitory effects of DMDS (Fig. 3d). Currently, due to technological limitations, genetic manipulation of mitochondrial proteins in vivo for a direct test of this hypothesis is not feasible. However, a similar phenomenon has been observed in the salmon louse (*Lepeophtheirus salmonis*), where resistance to the insecticide deltamethrin is prevalent, and COX is believed to be implicated[72]. In this case, a genetic analysis of mitochondrial haplotypes collected from various regions led to the identification of a crucial point mutation, specifically a Leu (L) to Ser (S) mutation at position 107 in COX subunit I (the primary catalytic site), among the resistant haplotypes[72]. Interestingly, this mutation nearly corresponds to one of the two amino acids (aa 108) in our hypothetical predictions.

When we tested our predictions based on these two amino acid sites by quantifying tolerance in 20 *Drosophila* species with varying combinations of amino acids at these sites, we observed an 85% alignment with the DMDS tolerance phenotype across species. However, three cases did not conform to our predictions: *D. mojavensis baja*, *D. merkatorum*, and *D. bipectinata*. We found that *D. mojavensis baja* (amino acids T & T) and *D. bipectinata* (amino acids S & T) were highly susceptible to DMDS in contrast to our predictions of being tolerant and intermediate species respectively. Further, *D. merkatorum* (amino acids S & T) was observed to tolerate DMDS contrary to our hypothesized prediction as an intermediate species. These contradictions suggest the potential involvement of other factors, such as contributions from subunits beyond the primary catalytic subunit or the participation of alternative channel mechanisms in this mode of toxicity. Additionally, we cannot exclude a hypothesis where the common ancestor of species from subgenera *Drosophila* and *Dorsilopha* possessed DMDS resistance where clade-specific losses led to the current observed resistance phenotypes. Moreover, an independent gain of resistance could also be hypothesized in the case of *D. ananassae*. Lastly, factors like size and body composition may also play a complementary role, necessitating further investigation to enhance our understanding of the complete mechanism underlying DMDS resistance.

Looking ahead, future technological advancements enabling genetic manipulation of mitochondrial DNA (mtDNA) would offer the means for a direct examination of our hypothesis.

In conclusion, we present *D. busckii* as the first species known to tolerate toxic DMDS where mitochondrial COX is likely to be involved. COX, in general, represents a significant target in the medical field and is emerging as a potential target for the development of novel insecticides. However, there is a dearth of experimental reports that delve into the functional mechanisms underlying COX inhibitor interactions to date[39,40,73]. Our study introduces a system and provides an opportunity to gain deeper insights into interactions involving mitochondria-mediated resistance.

Furthermore, we contemplate whether the evolution of short-chain oligosulfide preference and the ability to survive on DMDS have conferred advantages to *Dbus* as a species. Previous reports

suggest that *Dbus* may coexist alongside other cosmopolitan *Drosophila* species in complex ecological niches, such as garbage dumps in vegetable markets near human habitats[21,22]. In scenarios where a habitat offers a variety of substrates and is exploited by multiple species, drosophilids may exhibit spatial partitioning. Our experiments involving a species pair competing for egg-laying substrates clearly demonstrated niche separation between *Dbus* and *Dmel* when presented with their preferred substrates (spinach and oranges, respectively).

Moreover, the preference and tolerance for DMDS provide *Dbus* with a unique opportunity to identify and occupy an exclusive niche, where not many other drosophilid species can thrive and compete. In summary, our research sheds light on an intriguing case of evolution within the *Drosophila* genus and highlights the potential of this fascinating *Drosophila* species as a system for further exploration in the realm of evolution.

## Methods

### Ethics statement
This study on drosophilid flies was performed in Germany where the research on invertebrates does not require a permit from a committee that approves animal research.

### Drosophila stock
Multiple fly lines, including wild-type species and transgenic flies, were used in this study, and a detailed description of their original sources and stock numbers is listed in Supplementary Table 3. Flies were reared on different food media listed in Supplementary Table 4 and were maintained at 12:12 h light: dark cycle at 23 °C and 40% relative humidity.

### Chemical stimuli
All chemicals used in this study were purchased with the highest purity possible. A list of all odorants used along with their suppliers is available in Supplementary Table 1. Odorants were diluted in hexane for the single sensillum recording experiments to screen the *Dbus* antenna. Oligosulfides were diluted in mineral oil for conducting dose−response curve experiments. For the experiment explained in Fig. 2f, concentrations (v/v) were used as follows while mineral oil was used as a solvent. Methyl acetate ($10^{-2}$), 2-butanone ($10^{-2}$), acetone ($10^{-2}$), ethyl-3-hydroxybutyrate ($10^{-4}$), isopropyl benzoate ($10^{-4}$), DMDS ($10^{-2\cdot4}$). Both 2-butanone and acetone were pipetted freshly each time before puffing and were used only a single time per replicate. Sevoflurane was purchased from Sigma (CAS: 28523-86-6).

### Artificial substrate rotting
We developed and followed a standard protocol for artificially rotting substrates. Substrates were freshly purchased from the local supermarket, washed, and chopped into pieces of ~0.5 cm³ pieces. For rotting cauliflower, we used frozen and later thawed cauliflower pieces[25] and followed the same protocol as explained next. Cut substrates were immediately transferred into plastic containers of 20 ml volumetric capacity (https://www.aurosanshop.de/de/produktkategorien/laborbedarf/probenverarbeitung/proben-container/20ml-sample-container-white-cap-no-label-md-al-01980) and left open for ~24 h. at room temperature (RT). Caps were put on the containers on the following day in a manner such that an exchange of gases could take place, and these complete units were kept at RT for another 24 h. Finally, units were transferred to an incubator set at 32 °C overnight. Wilted substrate from the lowermost layer was used for subsequent experiments. A change in substrate color and texture, partial liquefaction, and odor change marked the generation of an artificially fermented substrate. Substrates were fermented in replicates simultaneously, eventually pooled together and random sampling was done for the final

experiments. The same protocol was followed for multiple substrates except for rotting potatoes, where they were obtained serendipitously from the supermarket. The complexity of the rotting process and varying rotting rates among substrates prompted us to use the term "relative oviposition preference" in all figures.

### Substrate chemical analysis (SPME-GC-MS)
Multiple substrates were tested in both fresh and artificially fermented conditions except for rotting potatoes, which were encountered serendipitously. Approximately 2 g of chopped substrate (~1.5 cm high in volume if measured from the vial bottom) was filled in 10 ml glass vials closed with a cap with polytetrafluoroethylene silicone septum and kept at 25 °C for at least 30 min to saturate the vial headspace with volatiles. The cap was penetrated with an SPME fiber coated with 100 μm polydimethylsiloxane (Supelco) and headspace volatiles were collected for 20 min at RT. The SPME fiber was injected directly into the inlet of a gas chromatograph machine (Agilent 5975) connected to MS and having a nonpolar HP5 column (Agilent 19091S-433U, 30 m length, 250 μm diameter, and 0.25 μm film thickness, Agilent Technologies) and helium as carrier gas. The temperature of the oven was held at 40 °C for 3 min, increased by 5 °C min⁻¹ to 280 °C. The final temperature was held for 5 min. The MS transfer line was held at 280 °C, the MS source at 230 °C, and the MS quad at 150 °C. Mass spectra were scanned in EMV mode in the range of 29 mz⁻¹ to 350 mz⁻¹. Chromatograms were visualized using Enhanced data analysis software (Agilent Chemstation, Agilent Technologies) and manually analyzed using NIST library 2.3. (https://chemdata.nist.gov). A principal component analysis of all chromatograms was generated by using an online software called XCMS version 3.7.1[74].

### Behavioral bioassays
Wild-type flies were used for oviposition and preference bioassays. Flies of both sexes were kept together for 3 days post-eclosion. Exactly 4-day-old females were used for behavioral studies. A group of 10 or 25 females with 3 or 5 males respectively was used for experiments conducted in salad boxes (transparent plastic boxes, ~5*7*~10 cm (w*l*h) with 10 ventilation holes punctured with forceps) or in larger BugDorm© cages of (~50 cm³, BugDorm-44545 F, https://shop.bugdorm.com/distributors.php). Flies were sorted one day before the experiment (3rd day) using CO₂ pads and supplied with yeast granules *ad libitum* overnight. Oviposition plate hardness was optimized by varying agarose concentration. Plates with 0.25% agarose concentration were used throughout the study due to ease of handling as well as due to the equal distribution pattern of egg laying (Supplementary Fig. 1b). A central hole was punctured in 0.25% agarose plates (Supplementary Fig. 1a) to make a cavity of ~8*9 mm (diameter * height). Stimuli were put in this cavity and covered with filter paper (Rotilabo-round filters, type 601 A, Carl Roth GmbH, Germany) of ~10 mm diameter. For experiments with whole substrates, a portion of a substrate in the appropriate stage was filled in the cavity while 10 μl of $10^{-2}$ odorants dissolved in mineral oil were used in the case of experiments using individual odorants. Two plates (test and control) were ~1 cm and 15 cm apart in salad boxes and BugDorm© cages respectively. No-choice experiments involved a presentation of a single substrate while binary choice experiments tested relative preference between two substrates or test odorant and mineral oil control. Experiments generally began around 1100 h and were terminated around the same time except for experiments involving testing fresh substrates. In the latter case, experiments began around 1700 h and were finished by 1100 h on a subsequent day (~18 h). Eggs were manually counted after 48 h with a 16 L: 8D photoperiod during testing. The oviposition index was calculated as $(T - C)/(T + C)$ where T represents the number of eggs on the test plate while C represents the same on the control plate. Trap assays

were used to test long-range preference towards substrates where traps were manually created by attaching pink paper cones on plastic vials (20 ml volume) containing rotting substrates (see "Artificial substrate rotting" section).

For assessing the co-existence of two species together (Fig. 1e), 5 females and 2 males of each species were mixed and kept together overnight in food vials supplemented with yeast granules, and the standard two-choice oviposition procedure was followed as described above. Here, the age of the flies was not controlled as different species reach sexual maturity at different ages. However, flies of no species were younger than 5 days.

## Toxicity assay

For survival experiments, two setups were used. The first setup consisted of normal fly food added with DMDS (Supplementary Fig. 5b1). Here, normal fly food (otherwise 0.4% in hardness; see Supplementary Table 4) was added with distilled water so as to reach a consistency of 0.25%. Food was melted and a calculated amount of pure DMDS was added to the melted food just before it started re-solidifying. Approximately 2 ml of the odorant mixed food was poured into small vials (7 * 10 mm, height * diameter) and was allowed to cool down and closed with a Styrofoam plug. 10 flies (>5-day old, mixed sexes) of each species were anesthetized on $CO_2$ pads and transferred into the odorant mixed food vials. Susceptibility (knock-down) was manually scored at one-hour time intervals. It was possible to confirm fly susceptibility by visual inspection. Yet, vials were inverted, tapped and live fly numbers were confirmed by checking negative gravitaxis.

Another setup was used in order to ensure the delivery of only volatiles from test compounds (Supplementary Fig. 5b2). The setup used here was adapted from an earlier report[37]. Here, 10 flies (>5-day old, mixed sexes) were transferred to 25 ml Falcon tubes and allowed to acclimatize for ~4 h. The main tube had a Styrofoam plug (~3 mm thick) at the end and just before the lid. Subsequently, a drop of 50 μl of the test compound was put in the lid, and the lid was closed. Fly paralysis was observed and recorded as described earlier. DMDS was dissolved in mineral oil while $NaN_3$ was dissolved in distilled water.

## Larval survival assay

A group of adults was kept on agarose plates with a central yeast dot as an oviposition stimulant. L1 larvae were observed within one day of egg laying and collected using a wet brush. For assessing larval survival on synthetic DMDS, 75 first instar larvae of either species were collected and placed on food containing either mineral oil or DMDS ($10^{-3}$). Developmental parameters were manually scored every day until adult (F1) emergence. Each species had ten replicates in each scenario (with or without DMDS), and the number of vials showing each developmental stage (e.g., L1, L2) was manually scored. It must be noted that Dbus larvae are surface feeders while Dmel larvae tend to dig into the food. Hence, in some instances, stage recording was partially not possible in case larvae feeding within the food.

## Whole substrate life cycle assay

Twenty, 4–6-day old flies (either Dbusk or Dmel) were briefly anesthetized on a $CO_2$ pad and transferred to vials (250 ml volume) filled with 25 g of rotting substrate (either mushroom, spinach or orange). These substrates were rotted using the artificial protocol mentioned earlier in the methods. A filter paper (3 * 10 cm) was placed in a vertical position touching the substrate in order to control substrate humidity and later provide a dryer yet course surface for pupation. The vials were kept at 22 °C, 70% RT until F1 adults were obtained.

## Sequence alignments

Available complete and partial sequences for Cytochrome Oxydase I of 327 species of the genus Drosophila and close relatives of the genus Zaprionus, Scaptomyza, Scaptodrosophila, Liodrosophila Stegana and Mycodrosophila were obtained from the National Center for Biotechnology of the National Library of Medicine of the National Institute of Health of the United States of America (https://www.ncbi.nlm.nih.gov). Sequences were aligned using Genious Prime v2023.2.1 (Biomatters Ltd.).

## Electrophysiology

Single sensillum recordings (SSR) were performed by following a protocol previously described in detail[75]. Generally, 5–10 days old female flies were used for the experiment. A single fly was gently pushed in a 200 μl pipette tip in a way that only half of the head was protruding out from the tip. The fly was held in the tip using laboratory wax. The antenna was supplementary using a glass capillary in order to expose either the medial or posterior side of the third antennal segment. A reference electrode was inserted in the eye while extracellular recordings from individual sensilla were performed using an electrochemically sharpened tungsten electrode. All odorants for the antennal screening experiment were diluted in hexane and tested at $10^{-4}$ conc. (v/v) unless stated otherwise. Oligosulfides were diluted in mineral oil for conducting dose–response experiments from the Bab2 sensillum in D. busckii. Diluted odorants were pipetted in an odor cartridge described previously[75] and the same cartridge was used not more than 3 to 5 times for dose–response and antenna screening experiments, respectively, unless stated otherwise.

## Statistical analysis

Statistical analyses were performed using GraphPad-Prism 9.1.1 (https://www.graphpad.com/scientific-software/prism/). SSR traces were analyzed using AutoSpike32 software 3.7 version (Syntech, NL 1998). Changes in action potential (spike count) were calculated by subtracting the number of spikes one second before (spontaneous activity) from those elicited one second after the onset of the stimulus. For behavioral data analyses, the raw data count was converted to an index. Such index replicates were first tested for normal (Gaussian) distribution using the Shapiro–Wilk normality test (significance = 0.05). Most of the data was observed to be normally distributed. For testing behavioral significance between two groups or between test and zero, an unpaired parametric t-test with Welch's correction was performed. Moreover, one sample t-test was used for data sets with identical values. For multiple comparisons between normally distributed groups, ordinary one-way ANOVA with multiple comparisons was performed. In the case of non-normal distribution, non-parametric ANOVA with the Kruskal–Wallis post hoc test was performed. Graphs were generated using GraphPad-Prism 9.1.1. and figures were constructed and processed with Adobe Illustrator CS5 and Adobe Photoshop (Adobe System Inc.).

## Reporting summary

Further information on research design is available in the Nature Portfolio Reporting Summary linked to this article.

# Data availability

All data are available in the main text or in the supplementary materials. Source data are provided as a source data file. NCBI Accession codes for COXI sequences used in the study are provided as Supplementary Data File 1. For any other data such as chromatograms or electrophysiology raw data files, correspondence and requests should be directed to B.S.H.

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

## Acknowledgements

We thank Dr. Marcus Stensmyr for his comments on the first version of the manuscript. Further, we thank Silke Trautheim, Roland Spiess, Ibrahim Alali and Manal Alali for their help with maintaining fly stocks. We also thank Regina Stieber, Angela Lehman and Kerstin Weniger for technical assistance and Swetlana Laubrich for administrative assistance. This study was supported by the Max Planck Society (to B.S.H and M.K) within the Max Planck Centre Next Generation Chemical Ecology and the International Max Planck Research School (IMPRS) at the Max Planck Institute of Chemical Ecology (to V.P.M). D.G. was supported by the Max Planck Society.

## Author contributions

All authors conceived the project. V.P.M., D.G., and B.S.H. designed the experiments. V.P.M. conducted experiments, made the figures, analyzed data, and wrote the first draft of the manuscript. D.G. conducted experiments and analyzed data for Fig. 4e. M.K. and B.S.H. reviewed and edited the first draft of the manuscript. All authors discussed the results and wrote the final version of the manuscript.

## Funding

## Competing interests

The authors declare no competing interests.
