## [Peer Review file · Nature Communications]

Preference for and resistance to a toxic sulfur volatile opens up a unique niche in *Drosophila busckii*

Corresponding Author: Professor Bill Hansson

Version 0:

Reviewer comments:

Reviewer #1

(Remarks to the Author)

In the current Manuscript entitled "Preference for and resistance to a toxic sulfur volatile open up a unique niche in *Drosophila busckii*", Mahadevan et al. investigate the interaction between the human commensal *Drosophila busckii* and (mainly) rotten vegetables as toxic hosts.

The authors perform a variety of behavioural experiments including dual choice egg laying assays, larval preference assays, resistance and eclosion assays to compare *D. busckii* to other fly species. Via chemical analysis they identify DMDS as a key chemical compound emitted by rotting vegetables, a preferred substrate for *Dbus* and (potentially) the "ab2B" homolog neuron as main responsive pathway with novel ligand specificity. Furthermore, they provide data that *Dbus* shows increased resistance to DMDS in survival assays and propose COX as a potential underlying genetic factor driving this resistance. Last, they find that *Dbus* larvae survive well on DMDS containing substrates, supporting tolerance to this chemical as critical step towards niche adaptation.

Overall, the provided work is sound and provides novel insights into host interaction in *Drosophila* flies. The work is original and advances the understanding of how flies might adapt to novel environments. The experiments are mainly (see proposals for revision below) well done and support the claims and cover a broad variety of biological phenomena. However, while I acknowledge that the performed experiments tackle several aspects of the interaction of *Dbus* with its host substrates, a more mechanistic investigations of the described phenomena would have made me even more excited about the presented manuscript. Nevertheless, I think this is important work and should be published. At several sections of the manuscript, the authors overstate their claims and I propose to temper or adapt their wording. Here a few suggestions:

Major

Part 1: host shift to short-chain oligosulfides

The phylogenetic tree in Figure 1b is little informative. I suggest introducing the species used in this study (similar detail as in Figure 3e) to put them into a phylogenetic context for the non-specialist reader.

A major portion of the text of the 'Results' first paragraph (lines 95-106) describes the oviposition substrate screening performed using a one-choice assay. This screening is crucial for the identification of potential *Dbus* favorite substrates, but the results are reported in Extended Figure 1. In my opinion, this makes it difficult for the reader to follow the flow of this part. I would suggest replacing panel Fig1c (little informative and methodological) with panel Extended Fig1c (and leave the cage schematics in the Extended figure).

Dbus shows an increased preference to oviposit close to DMDS emitting substrates only based on olfactory input. In the current presentation, however, it was not clear to me that the flies were not able to interact with the substrate in all assays. Just by reading the Material and Methods and by looking at Ext. Fig 1a, I eventually understood that only olfactory stimuli are presented to the fly, and they do not touch the substrate. I think this essential information should be made much clearer in the main text and probably also in Figure 1c. Moreover, while the authors introduce different assays in Figure 1c, it is not immediately evident which assay is used in which figure panel.

Lines 113-114: What is the difference between both assays? Only spatial scale?

Line 722: "For the preference index, traps were manually created by attaching pink paper cones on plastic vials (see artificially substrate rotting) containing rotting substrates." - What does this mean? When were trap assays used? Does this refer to Fig 1e? Here, "Relative Attraction preference" is a weird label and I would rather use "Olfactory Preference".

The authors use a small schematic in Figure 3e and it might be useful to add this to other Figure panels. Also, it would

facilitate reading if the axis labels for two-choice assays represented the two options (e.g. Fig 1i: choice between mineral oil and mineral oil + DMDS etc.).

Part 2: OSN characterization

Line 217-219: “In our recordings from *D. bus*, we identified eleven basiconic sensillum classes, out of which four were comparable to four *D. melanogaster* sensillum types (*Dmelab1*, *ab4*, *ab6*, and *ab9*). However, seven classes were novel and unique to *D. bus* (Extended fig. 3a-b).” This phrasing is confusing. The data indicates that for four sensilla types, homology to *Dmel* classes could be established with high confidence. For the other seven classes, homology was less clear. These are not “Novel and unique” classes. Later in the manuscript, the authors claim homology between *ab2B* neurons in ten *Drosophila* species but do not mention this here. They should adapt the text accordingly.

Line 236: “build on experience” sounds weird to me. Do the author want to say something like “most parsimonious”? Similarly, some phrasing in the subsequent sentences is inaccurate:

Lines 242: “transitioning directionally from the subgenus *Sophophora* to *Dorsilopha*, with the subgenus *Drosophila* as a transition zone (Fig. 2f & g).” This is not correct. Remove or rephrase. *D. immigrans* is not responding and from the presented data it is not clear if e.g. other *Dorsilopha* species gained DMDS responses or not.

Line 245: “and shed light on the evolution of this sensory trait across the *Drosophila* species”. Remove. The authors provide not really any data for this claim.

Overall, the authors provide convincing data in this section that *D. bus* is responding to DMDS and one neuron gets activated by this ligand. I wonder how the authors selected the species presented here but also in other Figures and what was the logic behind? E.g. *D. immigrans* seems to show increased egg laying rate on rotting substrate but no olfactory activity towards DMDS.

Is it known if the *ab2B* neuron plays a role in egg laying? Was it ever tested if activation of *ab2B* stimulates this behaviour in *Dmel* or do they authors think that this is a novel role of this pathway in *D. bus*? Of course, the best experiment would be to generate a loss of function mutant for the respective receptor in *D. bus* (but I acknowledge that this is probably out of the scope of the current study).

For the phylogenetical reconstruction of response profiles: Did the authors look at receptor sequences of the OR expressed in *ab2B* neurons, and could they identify residues involved in the observed ligand specificity change?

Part 3: Resistance

Here, the authors test the resistance of various fly species to DMDS. While it is clear which experiments were performed, I had issues following the order of presented data. It was weird to first read about “preliminary” experiments (Lines 269 to 289) looking at the survival of flies upon DMDS exposure and speculation about a potential role of mitochondria (“Therefore, such a reversible effect of DMDS in *Dmel* hinted towards a potential involvement of mitochondria in DMDS susceptibility”) followed by a clear statement about the role of DMDS in binding COX inhibiting ATP generation (“It is known that DMDS exerts its neurotoxic effect by non-competitively binding to the mitochondrial cytochrome c oxidase (COX), also known as complex IV, leading to the inhibition of ATP generation”). Would it not be much clearer to state this function of DMDS at the beginning of the section and then present Ext. Data. Fig 5? Potentially, parts of the data could be also moved to the main figure.

Figure 3b/c: “Data was collected in the form of the time point at which 100% mortality was observed and hence without SD. The green bar represents the concentration at which a drastic survival difference between tested species was observed and the corresponding concentration was chosen for subsequent experiments. No statistical analysis was performed as the data was absolute values of time when 100% mortality was observed.” Not clear to me. Was there just one replicate? Why is it not possible to present the data from multiple trials and the respective variation?

I acknowledge that it is difficult to establish a causal link between the identified amino acid changes in COX1 and the seen resistance phenotypes given the mitochondrial nature of these transcripts. I am, however, not fully convinced by the claimed link and I would like to see more evidence for this. Could the authors extend their analysis of COX1 activity (Figure 3h) to other species? Figure 3e clearly indicates that other species should behave like *D. bus*, *D. dana* or *Dmel* in this assay and this should be relatively easy to test.

The AOX overexpression results look very good.

Line 343-345: “In conclusion, our results strongly suggest the involvement of COX and differences in COX sensitivity as key factors contributing to the observed DMDS tolerance phenotypes across the tested species.” This statement is very strong and lacks direct evidence. Did the authors e.g. check expression levels of AOX across species? Given their overexpression data, maybe some species produce higher amount of this enzyme?

Minor

Colors

Authors apply the same color palette (orange vs green) to label different species (*Dmel* vs *D. bus*) as well as to label different vegetable substrates (orange vs spinach/potato/etc). I find this quite misleading; it could work for some figure panels (i.e. Figure 1d) but can cause confusion in others (Figure 1f, Figure 3e). I would suggest changing the color code for *D. bus* and *Dmel* and expanding the color palette for the schematics in Figure 3. Similarly, the use of ‘Darkened violin plots’ to report significant differences between groups in some figures (i.e. Figure 1d, Extended Figure 1c) could be changed to a more standard annotation (i.e. using asterisks like in Fig1h, Fig3h) to unify it throughout the manuscript.

Part 1: host shift to short-chain oligosulfides

Fig 1a: Please unify labelling: *D. bus* vs. *D. bus* later on

The cartoon representations of the vegetable substrates show fresh instead of rotten vegetable substrates. It might be worth changing this to better represent the substrates used.

Please update reference 25.

Some of the labels in the Figures are extremely small (e.g. Sup Fig 1a, “top view”; Sup Fig 1c, “n=7-15”): I would recommend

increasing size for better readability

Line 100: "In a no-choice assay, *Dbus* laid significantly more eggs on multiple substrates compared to the control (10 ul of distilled water), except for rotting potato, cucumber, and strawberry (Extended fig. 1c)." This comparison is not shown in the Figure or explained in the figure legend. Please add the respective analysis.

Figure 1f is missing x-axis title and color choice is confusing (see comment above).

Figure 1g: Where is DMS? Could the authors label this chemical in the chromatogram?

In Figure 1h the axis title 'Rotting spinach' is not very accurate. One could move the annotations '+DMDS' and '+DMTS' from the plot to the x-axis scale to fit the style of Fig 1i. In this panel some data are replotted from Figure 1d and this should be stated in the figure caption.

Where experiment reported in Fig 1j also performed for *Dmel*?

Ext. Data Fig 2: It is very difficult to identify columns throughout the whole heat-map. Could the authors introduce column separations to be able to distinguish different samples?

Part 2: OSN characterization

Ext. Data Fig 3: Heatmap: Ordering of ab1 neurons correct? (the green labelling in the supplementary table 2 has been lost). Did the authors use spike size for nomenclature or diagnostics? It might be worth clarifying this as the latter might have been more accurate to detect homology.

Figure 2b: From the x-axis is it not clear how many compounds show a high response. Is this only DMDS or are there multiple compounds in this peak?

Should the x-axis label in Fig 2c not rather read 'concentration (10-X v/v)'? (same for Extended Figure 3c).

The statistics is missing in Fig 2d.

Line 230: What is DPDS?

Part 3: Resistance

Line 271: "Pilot experiments": remove the pilot. These are final, presented results.

Line 747: "Appropriate controls were used and tested" What does this mean?

Ext. Data Fig 5: Please spell check: "*D. Melanogaster*"; 5d, x-axis is not in scale between left and right panels.

In Extended Figure 5f, x-axis is weirdly represented/mislabelled.

Reviewer #2

(Remarks to the Author)

Reviewer #3

(Remarks to the Author)

Pal Mahadevan et al. performed behavioral and electrophysiological experiments to demonstrate that *D. busckii* uses oligosulfides as olfactory cues to locate food sources and oviposition sites, which facilitates the colonization of new feeding and breeding niches in fermenting vegetables and fungi. Interestingly, DMDS, a volatile emitted by plants neurotoxic for several insect species, seems to be the main attractant and is detected by a specific class of olfactory sensory neurons. The author then showed that *D. busckii* was also resistant to DMDS at multiple life stages which might be due to a non-DMDS sensitive mitochondrial cytochrome C oxidase.

Overall, this study nicely provides a clear example of a niche adaptation shift resulting from a combination of attraction to and resistance against a toxic volatile.

Major comments:

Paragraph from l115 to 128: can you provide an actualized map of the distribution of the species you're mentioning? That will help the reader (especially a non-drosophilist specialist) to understand your niche adaptation hypothesis.

L166-168: It would be more relevant to show the larval attraction experiment along with Figure 4. It is not straightforward to link ovipository cues to larval attractants or even to larval beneficial food sources: the rotting process and the colonization of the site with different microorganisms will modify its chemical composition.

L222 paragraph: If I'm correct, you recorded the response of basiconics to DMDS only (cf extended fig 3.a). I would have expected to see responses to DMTS as well even if the rest of the manuscript is focused on DMDS only. Another sensillum can be involved in the oligosulfides detection, along with Bab2B.

L237: That paragraph may be difficult to follow for those not familiar with SSR recordings and OR responses. It would help to mention the bibliographic references that indicate that, in *D. melanogaster*, ab2B has been shown to respond to ethyl-3-hydroxybutyrate/ isopropyl benzoate and ab2A to methyl acetate/ 2-butanone. However, these responses are linked to the ORs expressed in these cells, and these genes could have been pseudogenized, or their function could have diverged over time. Additionally, a similar distribution pattern on the antenna does not necessarily indicate a clear homology between the cell types. As genomes are currently available for multiple drosophila species, including *Dbus*, designing FISH probes that target or59b and or85a (if they are conserved) and establishing the cartography of the cells expressing them would

strengthen the hypothesis that these "ab2-like" sensilla are indeed homologous.

L305: Did you look only at the COX1 sequences? If yes, is there a reason why? If not, can you mention the results for the other subunits?

L336: The number of replicates is too small to produce significant statistical results and draw "strong" conclusions on the COX activity differences. Would it be possible to increase the number of replicates?

L318-321/L507/L599/figure3e: Based on the amino-acid sequences and the resistant profiles of the species included, you also can't exclude the hypothesis that the ancestor of the *Drosophila/drosilopa* species was resistant to DMDS and that this resistance was multifactorial. Clade-specific events of DMDS-resistance losses or, in contrast, DMDS-resistance gain in *D. ananassae* could then have occurred independently.

Minor comments

Figure 1. 1a: You used the abbreviation Dbusk instead of Dbus, which is used in the rest of the manuscript. Can you also specify the size of the scale? Or just remove it?

1f: typo in *D. pseudoobscura*

1h-j please specify in the legend that these experiments are made with Dbus (even if it's logical)

l and j: the statistical tests are missing.

1g: it would have been interesting to have the profiles of the rotted strawberries here as well (as a control)

Extended fig 1

1C: please specify the legend for Dbus and Dmel for non-significant tests.

1e and f: specify mineral oil or T and C as in figure 1i.

Why 0.25% of agarose and not 0.125? please justify

L220: the prefix B has been added to all the Dbus sensillum classes, not only the 7 with divergent profiles.

Extended fig 3a: there is an issue with the name of the odorants

L229-230: please specify the DMS and DPDS abbreviation as they appear for the first time

L292: typo in the bibliography reference

Extended fig4: please indicate the divergence time

Version 1:

Reviewer comments:

Reviewer #1

(Remarks to the Author)

In their revised manuscript, Mahadevan et al. have addressed most of the questions I raised in my critiques of the original manuscript, which I appreciate. However, there are some remaining points that I would like the authors to clarify:

Major:

I still have an issue with the way mortality data was collected for Figures 3b,c (and Ext. Data Fig. 5e – is the x-axis label correct here?). Given that supporting evidence for a role of COX1 in DMDS resistance was removed in this revised version, these data should be presented in the best possible way. It is encouraging that authors used 7 replicates per time point. However, I do not follow why only the 100% mortality time point was recorded. One could assume that flies in 6 vials die after 1h, the ones in the 7th vial after 3h and this way of data representation would score 3h as time point of 100% mortality. As experiments were only scored once per hour, it is also not evident why Figure 3c depicts the first time point at 30min. Overall, it would be better representing the data if only 1h interval steps were indicated on the y-axis.

I ask the authors to repeat this experiment scoring the time of 100% mortality per vial and then acquire data for several independent replicates per condition per species. This will allow to statistically compare mortality between fly species at the given concentrations of DMDS and NaN3.

The authors state that they tried to de-orphanize DbusOr85a using the *D. melanogaster* empty neuron system but they did not find any ligand activating this receptor. I think this is an important result as it implies in light of experiments shown in Fig 2f that either pairing changed, another receptor is expressed, or (little likely) the empty neuron system is not suitable for characterizing Dbus receptors.

If the authors are confident about this result, I would strongly encourage them to show this data in the Supplement and discuss it briefly.

Minor:

Colour usage: The colour scheme of Fig. 1e is still confusing. The legend of this figure indicates "orange" to represent *D. melanogaster*, "green" *D. busckii*. However, this does not match the data shown in the Figure.

The same applies to Figure 3g and h where there is a mix of colours used in both panels what is confusing.

Larval preference: In my opinion it would merit mentioning in the context of Fig. 4a that *Dmel* flies were test ("We observed significantly less activity in the case of *Dmel* larvae when exposed to 10-2 v/v concentration of DMDS"). If they do not move, this would result in a neutral preference index. In the end, the comparative aspect of this work supports the specific niche adaptation of *Dbus*.

Ext. Data Figure 2: I appreciate that the authors introduced column separations in Ext. Data Figure 3 but Ext. Data Figure 2 would equally benefit from these separations.

Reviewer #2

(Remarks to the Author)

Reviewer #3

(Remarks to the Author)

The authors hav addressed the issues I raised in my previous review. I support the publication of the revised manuscript in Nature Communications.

Version 2:

Reviewer comments:

Reviewer #1

(Remarks to the Author)

The authors addressed all my concerns in the revised version of the manuscript. Congratulations!

REVIEWER COMMENTS

Reviewer #1 (Remarks to the Author):

In the current Manuscript entitled “Preference for and resistance to a toxic sulfur volatile open up a unique niche in *Drosophila busckii*”, Mahadevan et al. investigate the interaction between the human commensal *Drosophila busckii* and (mainly) rotten vegetables as toxic hosts. The authors perform a variety of behavioural experiments including dual choice egg laying assays, larval preference assays, resistance and eclosion assays to compare *D. busckii* to other fly species. Via chemical analysis they identify DMDS as a key chemical compound emitted by rotting vegetables, a preferred substrate for *Dbus* and (potentially) the “ab2B” homolog neuron as main responsive pathway with novel ligand specificity. Furthermore, they provide data that *Dbus* shows increased resistance to DMDS in survival assays and propose COX as a potential underlying genetic

factor driving this resistance. Last, they find that *Dbus* larvae survive well on DMDS containing substrates, supporting tolerance to this chemical as critical step towards niche adaptation. Overall, the provided work is sound and provides novel insights into host interaction in *Drosophila* flies. The work is original and advances the understanding of how flies might adapt to novel environments. The experiments are mainly (see proposals for revision below) well done and support the claims and cover a broad variety of biological phenomena. However, while I acknowledge that the performed experiments tackle several aspects of the interaction of *Dbus* with its host substrates, more mechanistic investigations of the described phenomena would have made me even more excited about the presented manuscript. Nevertheless, I think this is important work and should be published. At several sections of the manuscript, the authors overstate their claims and I propose to temper or adapt their wording. Here a few suggestions:

Response: We greatly appreciate the time and comments by the anonymous reviewer. We discuss below each specific concern and how we have addressed these with analyses in the current re-submission. We refer in our responses to the figure numbers and line numbering in the new submission to aid appreciation of where we have incorporated new data and text to respond to the reviewer’s concerns.

Major

Part 1: host shift to short-chain oligosulfides

The phylogenetic tree in Figure 1b is little informative. I suggest introducing the species used in this study (similar detail as in Figure 3e) to put them into a phylogenetic context for the non-specialist reader.

Response: We have changed figure 1b and have introduced the phylogeny of all species used in this study and is now figure 1a.

A major portion of the text of the 'Results' first paragraph (lines 95-106) describes the oviposition substrate screening performed using a one-choice assay. This screening is crucial for the identification of potential Dbus favorite substrates, but the results are reported in Extended Figure 1. In my opinion, this makes it difficult for the reader to follow the flow of this part. I would suggest replacing panel Fig1c (little informative and methodological) with panel Extended Fig1c (and leave the cage schematics in the Extended figure).

Response: We agree with the reviewer and have now replaced the methodological schematics figure panel with no-choice assay results previously shown in extended figure 1c (new figure 1b).

Dbus shows an increased preference to oviposit close to DMDS emitting substrates only based on olfactory input. In the current presentation, however, it was not clear to me that the flies were not able to interact with the substrate in all assays. Just by reading the Material and Methods and by looking at Ext. Fig 1a, I eventually understood that only olfactory stimuli are presented to the fly, and they do not touch the substrate. I think this essential information should be made much clearer in the main text and probably also in Figure 1c.

Response: We agree that this condition should be highlighted early in the text. We have now mentioned this condition clearly in the beginning of the results section (lines 115-119)

Moreover, while the authors introduce different assays in Figure 1c, it is not immediately evident which assay is used in which figure panel.

Response: We have now modified figure 1c by replacing it with no choice assay (previously extended figure 1c) and moved the bioassay schematics as extended figure 1c.

Lines 113-114: What is the difference between both assays? Only spatial scale?

Response: The assay conducted in extended figure 1d (line 146- originally line 113) was aimed to determine the oviposition preference while figure 1d (originally 1e and referred to in line 114) tested attraction behavior. The main difference between these two assays is at the spatial scale as the former were close-range assays (in a salad box) while the latter were conducted in a larger box (extended figure 1c). We have modified the phrasing in lines 146-148 to point out that trap assays were conducted in big cages.

Line 722: "For the preference index, traps were manually created by attaching pink paper cones on plastic vials (see artificially substrate rotting) containing rotting substrates." - What does this mean? When were trap assays used? Does this refer to Fig 1e?

Response: We have rephrased the sentence as "Trap assays were used to test long-range preference towards substrates where traps were manually created by attaching pink paper cones on plastic vials (20 ml volume) containing rotting substrates (see artificially substrate rotting)." lines 940-942. Trap assays were used in figure 1d (originally figure 1e)

Here, “Relative Attraction preference” is a weird label and I would rather use “Olfactory Preference”.

Response: We have changed the y-axis title to “relative olfactory preference”

The authors use a small schematic in Figure 3e and it might be useful to add this to other Figure panels.

Response: We have modified other figures for better readability.

Also, it would facilitate reading if the axis labels for two-choice assays represented the two options (e.g. Fig 1i: choice between mineral oil and mineral oil + DMDS etc.).

Response: We have modified figure 1h (originally figure 1i) by labelling the choice stimuli. We have also done the same in figure 4a (originally figure 1j).

Part 2: OSN characterization

Line 217-219: “In our recordings from *Dbus*, we identified eleven basiconic sensillum classes, out of which four were comparable to four *Dmel* sensillum types (*Dmelab1*, *ab4*, *ab6*, and *ab9*). However, seven classes were novel and unique to *Dbus* (Extended fig. 3a-b).” This phrasing is confusing. The data indicates that for four sensilla types, homology to *Dmel* classes could be established with high confidence. For the other seven classes, homology was less clear. These are not “Novel and unique” classes. Later in the manuscript, the authors claim homology between *ab2B* neurons in ten *Drosophila* species but do not mention this here. They should adapt the text accordingly.

Response: We agree with the reviewer and have modified the text as “However, we also encountered seven classes in *Dbus* where a clear homology to the known sensillum types in *Dmel* could not be directly established” (lines 293-295)

Line 236: “build on experience” sounds weird to me. Do the author want to say something like “most parsimonious”?

Response: We have removed “built on experience” phrase and simply replaced it with “next” (line 325)

Similarly, some phrasing in the subsequent sentences is inaccurate:

Lines 242: “transitioning directionally from the subgenus *Sophophora* to *Dorsilopha*, with the subgenus *Drosophila* as a transition zone (Fig. 2f & g).” This is not correct. Remove or rephrase. *D. immigrans* is not responding and from the presented data, it is not clear if e.g. other *Dorsilopha* species gained DMDS responses or not.

Response: We agree with the reviewer. We have removed this phrase.

Line 245: “and shed light on the evolution of this sensory trait across the *Drosophila* species”. Remove. The authors provide not really any data for this claim.

Response: We have removed this phrase.

Overall, the authors provide convincing data in this section that *Dbus* is responding to DMDS and one neuron gets activated by this ligand. I wonder how the authors selected the species presented here but also in other Figures and what was the logic behind? E.g. *D. immigrans* seems to show increased egg laying rate on rotting substrate but no olfactory activity towards DMDS.

Response: Our primary aim was to test several species representing different subgenera (*Sophophora* and *Drosophila*). Further, we selected representative species from well-known subgroups or radiations. E.g. species from the *ananassae* group, *obscura* group, *virilis-repleta* radiation, *mojavensis* group and *immigrans* radiation. We acknowledge that these species do not depict the complete picture of the *Drosophila* genus or for that matter of these individual groups themselves. However, at least some ecological background information about these species is available.

Is it known if the ab2B neuron plays a role in egg laying? Was it ever tested if activation of ab2B stimulates this behaviour in *Dmel* or do they authors think that this is a novel role of this pathway in *Dbus*?

Response: We tested oviposition behavior in *D. melanogaster* towards ethyl-3-hydroxy butyrate (one of the primary ligands for ab2B:Or85a). However, we observed neutral behavior when tested at 10^{-4} v/v concentration. However, it is still unclear if Or85a is expressed in *Dbus* and if it is involved in DMDS response. Our efforts at deorphanizing *Dbus*Or85a have not demonstrated its role in DMDS detection. Therefore, it could be speculated that a different OR could possibly be expressed in these OSNs resulting in a new pathway.

Of course, the best experiment would be to generate a loss of function mutant for the respective receptor in *Dbus* (but I acknowledge that this is probably out of the scope of the current study).

Response: We agree with the reviewer. One of the first steps to do so would be to identify the odorant receptor involved in DMDS detection in *Dbus*. However, as the receptor detecting DMDS is still unknown, generating a loss of function mutant would be a work to be done in the future.

For the phylogenetically reconstruction of response profiles: Did the authors look at receptor sequences of the OR expressed in ab2B neurons, and could they identify residues involved in the observed ligand specificity change?

Response: The phylogenetic reconstructions shown in the study are representative and based on earlier published reports. We indeed investigated if there is any evolutionary trend with respect to Or85a across the phylogeny. However, the amino acid sequence identity between *Dmel*Or85a and *Dbus*Or85a is very low (~40%). Although we proceeded to deorphanize, *Dbus*Or85a using the *D. melanogaster* empty neuron system, we were unable to identify any ligand (including DMDS) that would activate this *Dbus*Or85a. Therefore, the receptor involved in the detection of DMDS remains to be identified. Moreover, as DMDS is a molecule with substantially small molecular weight (and volume), it might prove to be challenging to predict its binding to various Or85a orthologs *in silico*.

Part 3: Resistance

Here, the authors test the resistance of various fly species to DMDS. While it is, clear which experiments were performed, I had issues following the order of presented data. It was weird to first read about “preliminary” experiments (Lines 269 to 289) looking at the survival of flies upon DMDS exposure and speculation about a potential role of mitochondria (“Therefore, such a reversible effect of DMDS in *Dmel* hinted towards a potential involvement of mitochondria in DMDS susceptibility”) followed by a clear statement about the role of DMDS in binding COX inhibiting ATP generation (“It is known that DMDS exerts its neurotoxic effect by non-competitively binding to the mitochondrial cytochrome c oxidase (COX), also known as complex IV, leading to the inhibition of ATP generation”). Would it not be much clearer to state this function of DMDS at the beginning of the section and then present Ext. Data. Fig 5? Potentially, parts of the data could be also moved to the main figure.

Response: We agree with the reviewer and now have modified the text and have begun the section starting with the role of mitochondria in DMDS susceptibility in lines 372-375.

Figure 3b/c: “Data was collected in the form of the time point at which 100% mortality was observed and hence without SD. The green bar represents the concentration at which a drastic survival difference between tested species was observed and the corresponding concentration was chosen for subsequent experiments. No statistical analysis was performed as the data was absolute values of time when 100% mortality was observed.” Not clear to me. Was there just one replicate? Why is it not possible to present the data from multiple trials and the respective variation?

Response: In this experiment, the data was pooled from at least 7 replicates. Here, data was collected at an hourly interval and the hour when 100% mortality was observed was recorded as the 100% knockdown hour. For instance, we observed 100% mortality in both *Dbus* and *Dmel* in all replicates when exposed to 10^{-2} v/v DMDS at $t=1$ hr. Therefore, the value is absolute 1 ($t = 1$ hr) for all 7 replicates and hence without any standard deviation. We have now also mentioned that the experiment was conducted only for 3 hours and have mentioned this as “Note that mortality was recorded only for 3 hours post exposure (dotted line) after which no mortality was

observed and data points corresponding to the dotted line represent no effect of the test compound on fly mortality.” (lines 485-488 & 494-496)

I acknowledge that it is difficult to establish a causal link between the identified amino acid changes in COX1 and the seen resistance phenotypes given the mitochondrial nature of these transcripts. I am, however, not fully convinced by the claimed link and I would like to see more evidence for this. Could the authors extend their analysis of COX1 activity (Figure 3h) to other species? Figure 3e clearly indicates that other species should behave like *D. busckii*, *D. ananassalis* or *D. melanogaster* in this assay and this should be relatively easy to test.

Response: First, we acknowledge that the first experimental set lacked sufficient sample size and had too high variability to allow definite conclusions. Therefore, we agree that additional replicates were warranted and we thank the reviewer for pointing this out. Consequently, we aimed to increase the number of replicates per species during the revision.

We performed these experiments once again with the same commercially available kits. We extended the COX1 analysis to three additional species including *D. mojavensis mojavensis*, *D. hydei* and *D. navojoa*. However, we realized that the kit reagents were highly sensitive and the reactions began even before the readings could take place resulting in high variation across replicates. This technical difficulty proved to be challenging despite our best efforts and it seems that this method is not suitable and reliable for testing such quick reactions.

Moreover, it was not possible to troubleshoot and standardize a new protocol in the available reviewing time due to logistical difficulties. However, we think that the remaining evidence from our manuscript clearly supports the notion that COX is involved in DMDS tolerance (figure 3g) and that species specific variation in COX protein architecture (figure 3c & e) could correspond to observed phenotypes.

However, as the nature of the methodology to test differences in COX activity turned out to be unreliable, we have removed this figure from the new version.

The AOX overexpression results look very good.

Response: We appreciate the positive comment.

Line 343-345: “In conclusion, our results strongly suggest the involvement of COX and differences in COX sensitivity as key factors contributing to the observed DMDS tolerance phenotypes across the tested species.” This statement is very strong and lacks direct evidence.

Response: We have removed this statement in the current version of the manuscript.

Did the authors e.g. check expression levels of AOX across species? Given their overexpression data, maybe some species produce higher amount of this enzyme?

Response: AOX is not expressed in insects and is naturally present in plants and some basal animals.

Minor

Colors

Authors apply the same color palette (orange vs green) to label different species (Dmel vs Dbus) as well as to label different vegetable substrates (orange vs spinach/potato/etc). I find this quite misleading; it could work for some figure panels (i.e. Figure 1d) but can cause confusion in others (Figure 1f, Figure 3e). I would suggest changing the color code for Dbus and Dmel and expanding the color palette for the schematics in Figure 3.

Response: We have modified the figures for better readability. We have labelled the substrates in figures 1 b-d for better clarity. Moreover, we have removed substrate illustrations from figure 1e and replaced them with titles to reduce confusion. As we did not use any substrates (e.g. orange) in the case of experiments shown in figure 3, we have not modified the figure.

Similarly, the use of 'Darkened violin plots' to report significant differences between groups in some figures (i.e. Figure 1d, Extended Figure 1c) could be changed to a more standard annotation (i.e. using asterisks like in Fig1h, Fig3h) to unify it throughout the manuscript.

Response: We have added asterisks to the "darkened violin plots" throughout the manuscript.

Part 1: host shift to short-chain oligosulfides

Fig 1a: Please unify labelling: Dbusk vs. Dbus later on

Response: We have changed *Dbusk* to *Dbus* in the labelling.

The cartoon representations of the vegetable substrates show fresh instead of rotten vegetable substrates. It might be worth changing this to better represent the substrates used.

Response: We intended to use these images as representative images of the substrates used and have modified the legends by mentioning that these substrates were used in rotting stages (e.g., lines 249-250)

Please update reference 25.

Response: We have updated the reference (new reference 26).

Some of the labels in the Figures are extremely small (e.g. Sup Fig 1a, "top view"; Sup Fig 1c, "n=7-15"): I would recommend increasing size for better readability

Response: We have modified these figures to ensure better readability.

Line 100: "In a no-choice assay, *Dbus* laid significantly more eggs on multiple substrates compared to the control (10 ul of distilled water), except for rotting potato, cucumber, and strawberry (Extended fig. 1c)." This comparison is not shown in the Figure or explained in the figure legend. Please add the respective analysis.

Response: We have mentioned this in the figure legends (lines 241-248) and have also modified the figure (figure 1b).

Figure 1f is missing x-axis title and color choice is confusing (see comment above).

Response: We have added x-axis title in the figure now. Thank you for pointing this out.

Figure 1g: Where is DMS? Could the authors label this chemical in the chromatogram?

Response: We found DMS only in the case of rotting spinach. We have modified the chromatogram by highlighting DMS in red.

In Figure 1h the axis title 'Rotting spinach' is not very accurate. One could move the annotations '+DMDS' and '+DMTS' from the plot to the x-axis scale to fit the style of Fig 1i. In this panel, some data are replotted from Figure 1d and this should be stated in the figure caption.

Response: We have modified the figure with proper axis titles. We appreciate the note about replotting the data. We have added a note in the figure legend (lines 276-278).

Where experiment reported in Fig 1j also performed for *Dmel*?

Response: No. We observed significantly less activity in the case of *Dmel* larvae when exposed to 10^{-2} v/v concentration of DMDS (as we eventually found out that *Dmel* larvae are susceptible to DMDS). The larval attraction assays were performed only in the case of *Dbus* larvae.

Ext. Data Fig 2: It is very difficult to identify columns throughout the whole heat-map. Could the authors introduce column separations to be able to distinguish different samples?

Response: Thank you for this comment and we agree about the difficulty in readability. We have modified this figure by introducing column separations.

Part 2: OSN characterization

Ext. Data Fig 3: Heatmap: Ordering of ab1 neurons correct? (the green labelling in the supplementary table 2 has been lost). Did the authors use spike size for nomenclature or diagnostics? It might be worth clarifying this, as the latter might have been more accurate to detect homology.

Response: Yes, the ordering and labelling in ab1 neurons is correct. We used diagnostic odors to detect homology. Surprisingly, the neuron with the largest amplitude (Bab1A neuron) responded to CO₂ as compared to ab1C neurons in reported *Dmel*. We have modified the green labelling and mentioned about the diagnostic odors criteria in the legend (lines 1532-1534).

Figure2b: From the x-axis is it not clear how many compounds show a high response. Is this only DMDS or are there multiple compounds in this peak?

Response: Thank you for pointing this out. This OSN type responded only to DMDS and the peak correspond only to DMDS. We have now removed x-axis lines and only have a single x-axis title.

Should the x-axis label in Fig2c not rather read 'concentration (10-X v/v)'? (same for Extended Figure 3c).

Response: We have modified the x-axis and corrected the title.

The statistics is missing in Fig2d.

Response: We have added the statistics in the figure.

Line 230: What is DPDS?

Response: DPDS stands for dipropyldisulfide. We have mentioned it in the text (line 315).

Part 3: Resistance

Line 271: "Pilot experiments": remove the pilot. These are final, presented results.

Response: We have removed the word pilot.

Line 747: "Appropriate controls were used and tested" What does this mean?

Response: We have modified this statement and rephrased it as "Appropriate controls (exposure to mineral oil and distilled water) were used and tested" We tested the survival of flies when exposed to the solvents. However, we did not observe mortality when exposed to either of them. However, the data overlapped with the survival curve in the case of *Dbus* and hence we have not shown the control data (lines 975-976).

Ext. Data Fig 5: Please spell check: "D. Melanogaster"; 5d, x-axis is not in scale between left and right panels.

Response: Thank you for pointing this out. We have corrected the spelling and modified the x-axis scale to match both panels in the extended figure 5d.

In Extended Figure5f, x-axis is weirdly represented/mislabelled.

Response: We have modified the x-axis in the extended figure 5f.

Reviewer #2 (Remarks to the Author):

Response: We greatly appreciate the time and comments by the anonymous co-reviewer.

Reviewer #3 (Remarks to the Author):

Pal Mahadevan et al. performed behavioral and electrophysiological experiments to demonstrate that *D. busckii* uses oligosulfides as olfactory cues to locate food sources and oviposition sites, which facilitates the colonization of new feeding and breeding niches in fermenting vegetables and fungi. Interestingly, DMDS, a volatile emitted by plants neurotoxic for several insect species, seems to be the main attractant and is detected by a specific class of olfactory sensory neurons. The author then showed that *D. busckii* was also resistant to DMDS at multiple life stages which might be due to a non-DMDS sensitive mitochondrial cytochrome C oxidase. Overall, this study nicely provides a clear example of a niche adaptation shift resulting from a combination of attraction to and resistance against a toxic volatile.

Response: We greatly appreciate the time and comments by the anonymous reviewer. We discuss below each of their specific concerns and how we have addressed these with analyses, where possible, in the current re-submission. We refer in our responses to the figure and lines numbering in the new submission to aid appreciation of where we have incorporated new data and text to respond to the reviewers' concerns.

Major comments:

Paragraph from l115 to 128: can you provide an actualized map of the distribution of the species you're mentioning? That will help the reader (especially a non-drosophilist specialist) to understand your niche adaptation hypothesis.

Response: We have modified figure 1a by depicting the phylogenetic relationship between all the species tested in this study and have also referred to the extended figure 5h map concerning the geographical distribution of those.

L166-168: It would be more relevant to show the larval attraction experiment along with Figure 4. It is not straightforward to link ovipository cues to larval attractants or even to larval beneficial food sources: the rotting process and the colonization of the site with different microorganisms will modify its chemical composition.

Response: We agree with the reviewer. We have moved the larval attraction experiment to figure 4 along with the rest of the experiments concerning larvae.

L222 paragraph: If I'm correct, you recorded the response of basicoinics to DMDS only (cf extended fig 3.a). I would have expected to see responses to DMTS as well even if the rest of the manuscript is focused on DMDS only. Another sensillum can be involved in the oligosulfides detection, along with Bab2B.

Response: Yes, the screening panel used to screen all basicoinic sensillum types in *D. busckii* only had DMDS as a short-chain oligosulfide representative. Although we cannot completely rule out the possibility that another sensillum class might be involved in DMTS detection, the chances seem low. We observed a significant shift in sensitivity in the case of Bab2B type when compared between responses to DMDS and DMTS (less sensitive to DMTS). Moreover, both DMDS and DMTS belong to the short-chain oligosulfide chemical class. The OSN classes belonging to the remaining 10 sensillum types (identified from our antennal screening, extended figure 3a) were observed to respond to different chemical classes (such as C9 compounds or esters) but not to DMDS. Hence, it can be speculated that Bab2B could be the primary OSN type involved in the detection of short-chain oligosulfides.

L237: That paragraph may be difficult to follow for those not familiar with SSR recordings and OR responses. It would help to mention the bibliographic references that indicate that, in *D. melanogaster*, ab2B has been shown to respond to ethyl-3-hydroxybutyrate/ isopropyl benzoate and ab2A to methyl acetate/ 2-butanone. However, these responses are linked to the ORs expressed in these cells, and these genes could have been pseudogenized, or their function could have diverged over time. Additionally, a similar distribution pattern on the antenna does not necessarily indicate a clear homology between the cell types. As genomes are currently available for multiple drosophila species, including *D. busckii*, designing FISH probes that target *or59b* and

or85a (if they are conserved) and establishing the cartography of the cells expressing them would strengthen the hypothesis that these "ab2-like" sensilla are indeed homologous.

Response: We have added the bibliographic references regarding known literature in *Dmel* with respect to the OSN response profile (lines 321-324). We agree with the reviewer regarding the comments about potential changes at the receptor level. However, significantly less information is available about the receptors and their expression in *Dbus* compared to other species. Although it would be interesting to investigate the receptors involved in DMDS detection in *Dbus*, establishing homology based on *in-situ* hybridization and investigating molecular evolution across several species is currently out of the scope of the present study. Moreover, it is also possible that a new receptor (other than Or85a) is expressed in such a location. Lastly, we agree that a similar distribution pattern on the antenna does not indicate homology. However, we referred to the region to facilitate future investigations and to point out towards conserved distribution of big basiconic sensilla in the proximal regions.

L305: Did you look only at the COXI sequences? If yes, is there a reason why? If not, can you mentioned the results for the other subunits?

Response: We focused mainly on COXI as most of the literature points towards the involvement of this subunit in electron transport^{1,2}. The COX enzyme comprises 14 subunits (only I-III involved in catalytic activity). We compared the sequences of both COX II and III across several species with different phenotypes. However, the sequences were highly conserved thereby predicting and correlating the involvement of amino acids with the observed resistance phenotype was not possible.

L336: The number of replicates is too small to produce significant statistical results and draw "strong" conclusions on the COX activity differences. Would it be possible to increase the number of replicates?

Response: First, we acknowledge that the first experimental set lacked sufficient sample size and had too high variability to allow definite conclusions. Therefore, we agree that additional replicates were warranted and we thank the reviewer for pointing this out. Consequently, we aimed to increase the number of replicates per species during the revision.

We performed these experiments once again with the same commercially available kits. We extended the COX1 analysis to three additional species including *D. mojavensis mojavensis*, *D. hydei* and *D. navojoa*. However, we realized that the kit reagents were highly sensitive and the reactions began even before the readings could take place resulting in high variation across replicates. This technical difficulty proved to be challenging despite our best efforts and it seems that this method is not suitable and reliable for testing such quick reactions.

Moreover, it was not possible to troubleshoot and standardize a new protocol in the available reviewing time due to logistical difficulties. However, we think that the remaining evidence from our manuscript clearly supports the notion that COX is involved in DMDS tolerance (figure 3g) and that species specific variation in COX protein architecture (figure 3c & e) could correspond to observed phenotypes.

However, as the nature of the methodology to test differences in COX activity turned out to be unreliable, we have removed this figure from the new version.

L318-321/L507/L599/figure3e: Based on the amino-acid sequences and the resistant profiles of the species included, you also can't exclude the hypothesis that the ancestor of the drosophila/Dorsilopa species was resistant to DMDS and that this resistance was multifactorial. Clade-specific events of DMDS-resistance losses or, in contrast, DMDS-resistance gain in *D. ananassae* could then have occurred independently.

Response: We agree with the reviewer that the current observed phenotypic pattern could be a result of loss of function across several clades as well as convergent evolution in the case of *D. ananassae* (a different subgenus). We have mentioned this hypothesis in the discussion in the present version (lines 806-809).

Minor comments

Figure 1. 1a: You used the abbreviation Dbusk instead of Dbus, which is used in the rest of the manuscript. Can you also specify the size of the scale? Or just remove it?

Response: Thank you for pointing this out. We have corrected the same and removed the scale.

1f: typo in *D. pseudoobscura*

Response: Thank you for pointing this out. We have corrected the same.

1h-j please specify in the legend that these experiments are made with Dbus (even if it's logical)

Response: We have modified the legend by adding *Dbus* as a species tested (lines 273 & 280).

l and j: the statistical tests are missing.

Response: We have added the statistical tests to figures 1l & j (now figures 1h & 4a).

1g: it would have been interesting to have the profiles of the rotted strawberries here as well (as a control)

Response: We did not test the headspace of rotting strawberries as all of our experiments included rotting oranges. However, previous reports have shown a dominance of esters in the headspace of strawberries in a rotting state³.

Extended fig 1

1C: please specify the legend for Dbus and Dmel for non-significant tests.

Response: We have specified this difference in the figure legend (now figure 1b, lines 246-249).

1e and f: specify mineral oil or T and C as in figure 1i.

Response: We have added the appropriate choices in the modified figures.

Why 0.25% of agarose and not 0.125? Please justify

Response: We found that both 0.25% and 0.125% agarose elicited similar oviposition behavior (and egg counts). However, 0.125% agarose plates were more difficult to handle (due to the semisolid nature of the low-concentration agarose) during transport and egg counting. Lastly, our experiments involved puncturing a hole in the agarose in order to present a rotting substrate. The puncturing process was significantly more delicate with lower concentration and hence, 0.25% was chosen for further experiments. We have mentioned this in lines 923-926.

L220: the prefix B has been added to all the Dbus sensillum classes, not only the 7 with divergent profiles.

Response: We have rephrased the sentence and assigned the prefix "B" to all sensillum classes in *Dbus*" (line 296)

Extended fig 3a: there is an issue with the name of the odorants

Response: We have corrected the odorant legends in the figure.

L229-230: please specify the DMS and DPDS abbreviation as they appear for the first time

Response: We have added the full form of those abbreviations in the text (line 315).

L292: typo in the bibliography reference

Response: We have corrected the reference.

Extended fig4: please indicate the divergence time

Response: We have added the divergence time to the figure.

REVIEWER COMMENTS

Reviewer #1 (Remarks to the Author):

In their revised manuscript, Mahadevan et al. have addressed most of the questions I raised in my critiques of the original manuscript, which I appreciate. However, there are some remaining points that I would like the authors to clarify:

Major:

I still have an issue with the way mortality data was collected for Figures 3b,c (and Ext. Data Fig. 5e – is the x-axis label correct here?). Given that supporting evidence for a role of COX1 in DMDS resistance was removed in this revised version, these data should be presented in the best possible way. It is encouraging that authors used 7 replicates per time point. However, I do not follow why only the 100% mortality time point was recorded. One could assume that flies in 6 vials die after 1h, the ones in the 7th vial after 3h and this way of data representation would score 3h as time point of 100% mortality.

As experiments were only scored once per hour, it is also not evident why Figure 3c depicts the first time point at 30min. Overall, it would be better representing the data if only 1h interval steps were indicated on the y-axis.

I ask the authors to repeat this experiment scoring the time of 100% mortality per vial and then acquire data for several independent replicates per condition per species. This will allow to statistically compare mortality between fly species at the given concentrations of DMDS and NaN₃.

Response: We appreciate the comments on this experiment and agree with the representation of the data. We have modified figures 3b&c with appropriate axis labels and added appropriate statistical analysis. Further, we have demonstrated mortality only at one concentration (3×10^{-3} v/v) in the main figure as we have used this concentration in the subsequent experiments. Tolerance results at all the remaining concentrations to either DMDS or NaN₃ are now shown in the extended figure 5d. Moreover, both species showed significantly lower tolerance to NaN₃ compared to DMDS and hence, observations were recorded in intervals of 30 mins. We thank the reviewer for pointing out these issues and the faulty x-axis label in the case of the sevoflurane experiment. We have corrected the label.

The authors state that they tried to de-orphanize DbusOr85a using the *D. melanogaster* empty neuron system but they did not find any ligand activating this receptor. I think this is an important result as it implies in light of experiments shown in Fig 2f that either pairing changed, another receptor is expressed, or (little likely) the empty neuron system is not suitable for characterizing Dbus receptors.

If the authors are confident about this result, I would strongly encourage them to show this data in the Supplement and discuss it briefly.

Response: We agree with the reviewer regarding the importance of this result. However, we observed extremely low spontaneous activity in the OSNs heterologously expressing the OR (DbusOr85a) in the *Dmel* empty neuron system (the system reported in Chahda et al., 2019). Moreover, the spontaneous activity was not present at all in some tested flies (although the expression was confirmed later using PCR). Therefore, we decided not to add this dataset to the current version of the manuscript.

Minor:

Colour usage: The colour scheme of Fig. 1e is still confusing. The legend of this figure indicates “orange” to represent *D. melanogaster*, “green” *D. busckii*. However, this does not match the data shown in the Figure.

Response: We thank the reviewer for pointing this out. We have corrected the figure legend in Fig. 1e.

The same applies to Figure 3g and h where there is a mix of colours used in both panels what is confusing.

Response: We have now changed the figure colors in both figures 3g and h.

Larval preference: In my opinion it would merit mentioning in the context of Fig. 4a that *Dmel* flies were test (“We observed significantly less activity in the case of *Dmel* larvae when exposed to 10⁻² v/v concentration of DMDS”). If they do not move, this would result in a neutral preference index. In the end, the comparative aspect of this work supports the specific niche adaptation of *Dbus*.

Response: We have now mentioned that *Dmel* flies were tested by exposure to DMDS in the same context of Fig. 4a in lines 411-412.

Ext. Data Figure 2: I appreciate that the authors introduced column separations in Ext. Data Figure 3 but Ext. Data Figure 2 would equally benefit from these separations.

Response: We indeed checked the possibility of adding column separations in the extended data figure 3. However, as we are not listing row names (i.e., individual compounds) these added spaces did not ultimately enhance the overall figure meaning. We are of course open to sharing the raw data (chemical analysis of various stimuli) on request.

Reviewer #2 (Remarks to the Author):

Response: We greatly appreciate the time and comments by the anonymous co-reviewer.

Reviewer #3 (Remarks to the Author):

The authors have addressed the issues I raised in my previous review. I support the publication of the revised manuscript in Nature Communications.

Response: We are grateful for the time and suggestions from the reviewer as they have greatly helped improve the original version of the manuscript. We thank the reviewer for the positive comments and the review decision.